# Agricultural Conservation Practices and Aquatic Ecological Responses

**Richard E. Lizotte** [1,*] , **Peter C. Smiley** [2] , **Robert B. Gillespie** [3] **and Scott S. Knight** [4]

1 USDA-ARS National Sedimentation Laboratory, Oxford, MS 38655, USA
2 USDA-ARS Soil Drainage Research Unit, Columbus, OH 43210, USA; Rocky.Smiley@ars.usda.gov
3 Department of Biology, Purdue University Fort Wayne, Fort Wayne, IN 46805, USA; gillespi@pfw.edu
4 University of Mississippi Field Station, The University of Mississippi, Abbeville, MS 38601, USA; sknight@olemiss.edu
* Correspondence: Richard.Lizotte@USDA.gov; Tel.: +1-66-2232-2956

**Abstract:** Conservation agriculture practices (CAs) have been internationally promoted and used for decades to enhance soil health and mitigate soil loss. An additional benefit of CAs has been mitigation of agricultural runoff impacts on aquatic ecosystems. Countries across the globe have agricultural agencies that provide programs for farmers to implement a variety of CAs. Increasingly there is a need to demonstrate that CAs can provide ecological improvements in aquatic ecosystems. Growing global concerns of lost habitat, biodiversity, and ecosystem services, increased eutrophication and associated harmful algal blooms are expected to intensify with increasing global populations and changing climate. We conducted a literature review identifying 88 studies linking CAs to aquatic ecological responses since 2000. Most studies were conducted in North America (78%), primarily the United States (73%), within the framework of the USDA Conservation Effects Assessment Project. Identified studies most frequently documented macroinvertebrate (31%), fish (28%), and algal (20%) responses to riparian (29%), wetland (18%), or combinations (32%) of CAs and/or responses to eutrophication (27%) and pesticide contamination (23%). Notable research gaps include better understanding of biogeochemistry with CAs, quantitative links between varying CAs and ecological responses, and linkages of CAs with aquatic ecosystem structure and function.

**Keywords:** conservation; ecology; habitat; eutrophication; pesticides; agroecosystems





## 1. Introduction

Proper management of water resources including water quality and water quantity in agricultural watersheds is a key component to maintaining healthy aquatic ecosystems. Healthy aquatic ecosystems are sustainable ecosystems that exhibit resilience in their structure (i.e., biodiversity) and function (i.e., organic matter processing) in response to external stress [1] and subsequently able to provide a variety of ecosystem services including clean water, climate regulation, habitat for plants and animals, nutrient cycling, and productivity [2–4]. Ecosystem management approaches that focus on maximizing one ecosystem service result in declines of biodiversity and other ecosystem services [5] Agricultural land use within agricultural watersheds has impacted the ecosystem structure and function of lentic and lotic ecosystems via altered hydrology and increased erosion as a result of land use change and channelization and increased pollution resulting from excess inputs of nutrients, pesticides, and other agricultural contaminants [6]. Subsequently, there is an interest in reducing the impacts of agriculture on aquatic ecosystem structure and function with the use of conservation agriculture practices (CAs) that have been widely implemented in developed regions of the world including Europe, North America, Asia, and Australia [7–10] through various agricultural agency conservation programs. However, developing countries often have very low CA implementation that greatly limits effective water resource management [11]. Additionally, most agricultural land within the developed

regions of the world is privately owned and subsequently the approach towards managing agricultural watersheds differs from that of public wilderness tracts. The management of agricultural watersheds focuses on cropland production and addressing the subsequent water quality issues via the voluntary adoption by individual landowners of CAs and management of public wilderness tracts focuses on conservation via management by government agencies [3,12].

The agricultural community traditionally has viewed CAs as methods for managing soil and water resources and to improve agricultural production. Many of these same practices are increasingly being viewed as practices that are capable of mitigating environmental impacts of agriculture on aquatic ecosystems. There are numerous publications that provide information on the results of field and plot scale studies examining the physical, chemical, and/or hydrological impacts of conservation practices (see reviews by [13–18]). Unfortunately, information on the ecological effects of agricultural CAs on aquatic ecosystems at larger spatial scales such as sub-watershed to watershed levels is still limited. Ecological effects are typically documented with field research involving the assessment of changes in biota, physical habitat variables, and/or chemical variables that represent the responses of ecosystem structure and/or function to the implementation of CAs and the assessment of stressor response relationships between the biota, physical habitat variables, or chemical variables with selected ecosystem stressors [19,20]. In an attempt to address the limited information on the effects of CAs on aquatic ecosystems at sub-watershed and watershed scales, several countries have enlisted the resources of government agencies including, but not limited to, The French Ministry of Agriculture and Food, Agriculture and Agri Food Canada, Chinese Ministry of Agriculture, Australia Department of Agriculture and Food, New Zealand Ministry for Business, Innovation and Employment's—Clean Water, and The United States Department of Agriculture (USDA) [21–25]. The French Ministry of Agriculture and Food, Territoires d'Innovation projects are agroecosystem living laboratory approaches to enhance innovation in sustainability and resilience to protect soil and biodiversity in agricultural watersheds [21]. Similarly, AAFC also utilizes an agroecosystem living laboratory approach to address agriculturally sourced environmental issues affecting soil and water management and biodiversity with a changing climate. The AAFC living labs projects allow rapid adoption of sustainable practices through close collaboration among researchers, stakeholders, and land-use managers (e.g., farmers) [21]. Within China, Chinese Nationally Important Agricultural Heritage Systems (China—NIAHS) are selected agricultural systems that demonstrate long-term (at least 100 years) sustainable historic agricultural practices providing resilience to extreme conditions (e.g., drought, flooding). These valuable heritage systems provide valuable lessons for biodiversity conservation, soil and water conservation, climate regulation, and nutrient cycling while providing food and livelihood security for the rural community [22]. In Australia, Townsend et al. [23] assessed economic valuation of multiple ecosystem services through tradeoffs between reforestation and agricultural land use as payments for ecosystem services (PES) in a process called 'bundling'. Ecosystem services via reforestation could include water conservation, carbon sequestration, eco-tourism, and conservation of biodiversity. Townsend et al. [23] suggested that the greatest likelihood for success of such a program would need to be through government establishing appropriate mechanisms to subsidize PES payments for water quality improvement. McDowell et al. [24] discussed in detail New Zealand's water quality policy as outlined in the country's National Policy Statement on Freshwater Management that requires integrated and sustainable water resource management. New Zealand has a combination of mandatory regulation and voluntary initiatives coinciding with monitoring and evaluation assessment programs to demonstrate successful implementation and regulatory compliance. By comparison, the United Kingdom (UK) has many similar policies and programs as New Zealand. However, unlike New Zealand, the UK utilizes subsidy payments or financial incentives in addition to the previous policies and programs to further encourage stakeholders and land managers to investment in conservation agriculture practices. The USDA initiated the Conservation Effects Assessment

Project (CEAP) as a multi-agency collaboration of the Natural Resources Conservation Service (NRCS), Agricultural Research Service (ARS), and National Institute of Food and Agriculture [25]. One of the goals of CEAP is to quantify the effects of CAs at the sub-watershed and watershed scales and to develop sound science for managing agricultural watersheds to maintain or improve ecosystem services [25,26]. As of 2019, CEAP has 23 active watershed projects (Figure 1). The USDA ARS's Watershed Assessment Study consists of in-depth studies conducted within 14 watersheds in the United States to provide information on the effects of conservation practices at the watershed scale that will assist with the National Assessment's modeling efforts.

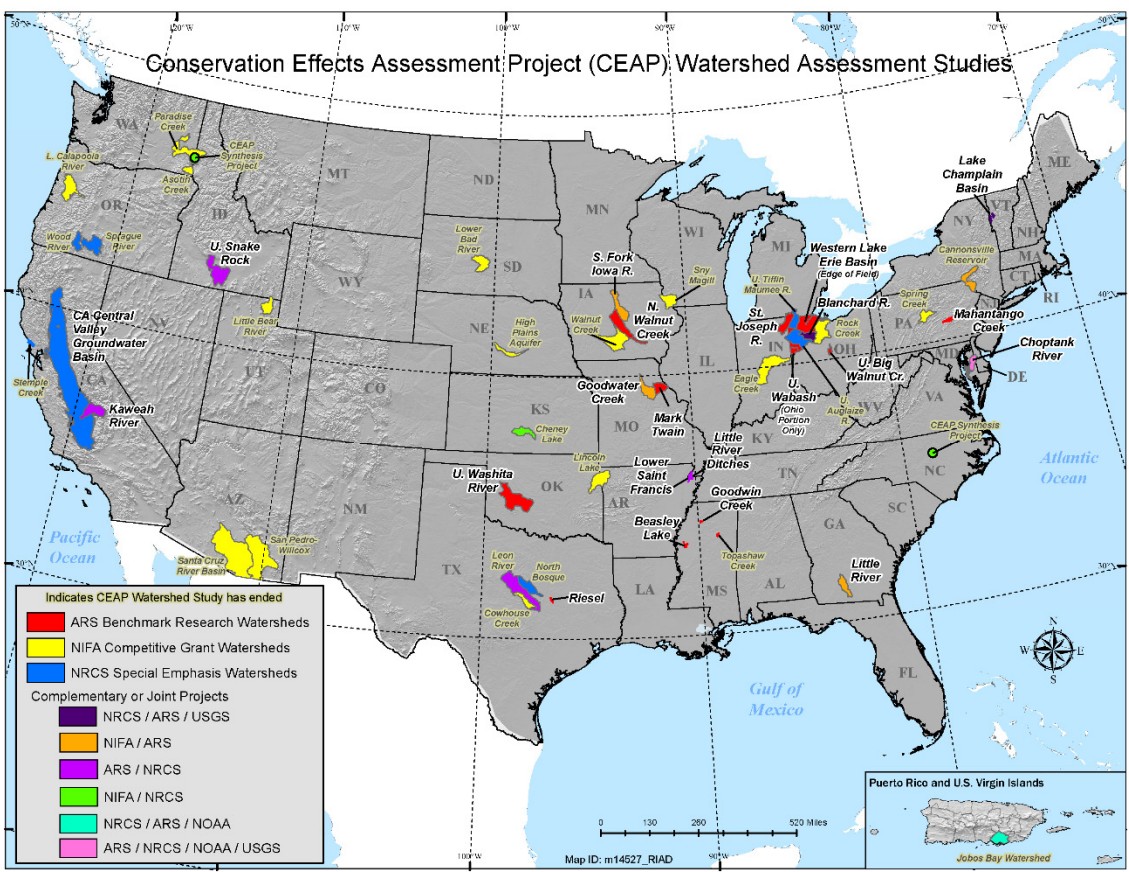

**Figure 1.** Map of the United States Department of Agriculture, Natural Resources Conservation Service, Conservation Effects Assessment Project Watershed Assessment study watersheds within the United States. The map is available at https://www.nrcs.usda.gov/wps/portal/nrcs/main/national/technical/nra/ceap/ws/ (accessed on 20 March 2021).

These 14 watersheds were selected because they were sites involved in long-term watershed research that was anticipated to continue in the future. By comparison, Agriculture and Agri Food Canada, Agriculture and Environment, Living Laboratories Initiative has a similar focus to France's Territoires d'Innovation and USDA's CEAP in addressing sustainable agriculture and agri-environmental issues [21]. Canada's Living Laboratories Initiative goals include working with farmers and producers to mitigate water contamination, improve soil and water conservation, and enhance habitat and biodiversity in agricultural watersheds.

Thus, ecological research conducted in conjunction with CA research efforts can provide information on the ecological effects of CAs at broad scales. Our objective is to summarize and synthesize research conducted to quantify ecological responses to CAs since 2000. We will also identify critical ecological data gaps and future research needs to better assess how agricultural conservation practices may improve aquatic ecosystem services.

## 2. Methods

Identification and collection of studies reported globally in this synthesis included peer-reviewed journal articles and non-peer reviewed book chapters, government reports, student graduate theses, and proceedings papers published from years 2000–2020 (Figure 2). These were obtained through web-based search engines including SCOPUS, Google Scholar, BioOne, and the US Department of Agriculture National Agricultural Library (NAL). Because the focus of the synthesis is on aquatic ecosystems, initial keyword and syntax used for the literature search included this focus: stream, river, lake, and pond.

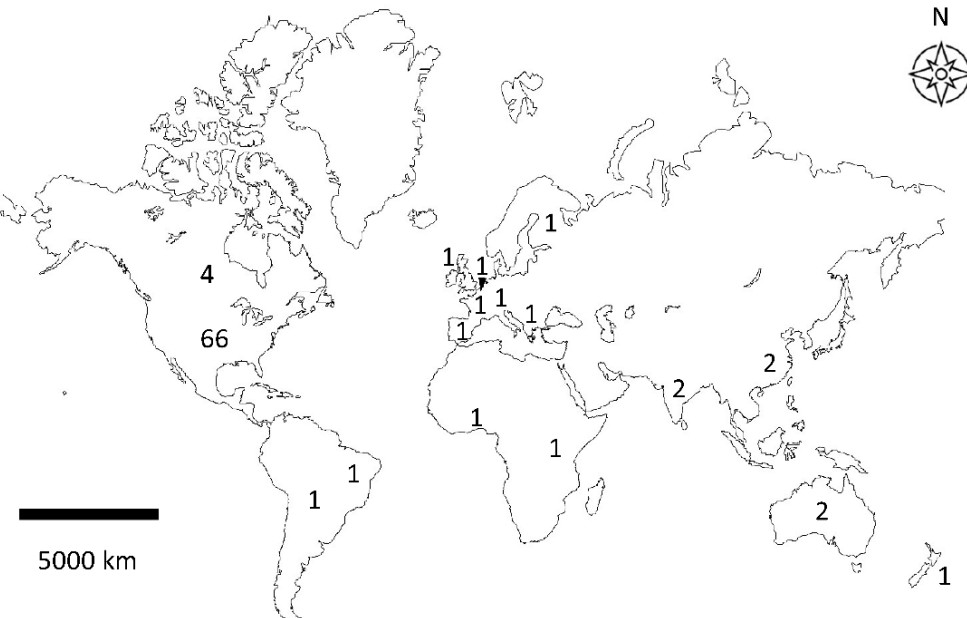

**Figure 2.** World map of the number of studies assessed from different countries from a delineated literature search for aquatic ecological responses to conservation agriculture practices from 2000–2020 [13–20,27–111].

Keywords and syntax used to delineate the literature search for CAs included: (a) conservation agriculture; (b) agricultural conservation practices; (c) best management practices; (d) vegetated buffers; (e) riparia (n); (f) wetland; (g) bioreactor; and (h) integrated pest management. Ecological structure keywords and syntax used in conjunction with CA variables above to further refine the search included: (a) biodiversity; (b) habitat; (c) community; (d) assemblage; (e) fish; (f) invertebrate; (g) amphibian; (h) mammal; (i) algae; and (j) plant. Ecological function keywords and syntax used in conjunction with above CA variables to further refine the search included: (a) nutrients (nitrogen, phosphorus); (b) carbon; (c) pesticides; (d) toxicity; (e) eutrophication; (f) brownification; and (g) denitrification. Using these criteria, 114 studies were selected for further assessment to ascertain the absence or presence of qualitative or quantitative links between measured aquatic ecological response variables and CAs within the study. After this second selection stage, 88 studies from 2000–2020 were selected for summary analysis (Figure 3). No published studies were observed for the years 2000 and 2018 while the most published studies occurred for the years 2008 and 2012 with 10 and 13 publications, respectively.

Information collected from the previously described literature search was tabulated and summarized using excel spreadsheets that included the continent, country, CA practice, ecological structure, ecological function, the ecological response to CAs (positive, negative, neutral, unknown), and the study authors. Descriptive and summary statistics were used to synthesize the data and determine the frequency of occurrence for the above listed variables. The tabulated information was used to discern where studies are being conducted globally, what CAs are being assessed, and what ecological responses are being measured.

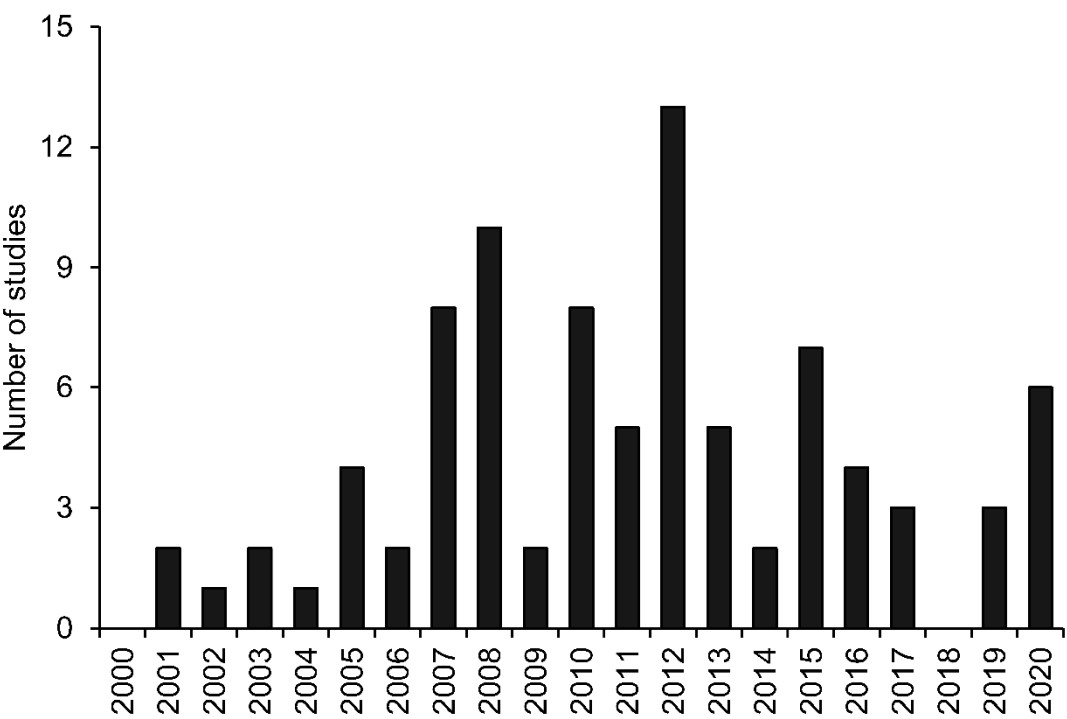

**Figure 3.** Number of studies assessed from delineated literature search for aquatic ecological responses to conservation agriculture practices from 2000–2020 [13–20,27–111].

## 3. Results and Discussion

### 3.1. Study Locations and Conservation Agriculture Practices

The synthesis of the literature from 2000–2020 delineated that research on aquatic ecological responses to CAs occurred on six continents within 17 countries [13–20,27–111]. Most studies were reported between 2007 and 2015 with 2018 being the only year without at least one report (Figures 2 and 3). Approximately 78% of all published studies were conducted in North America and primarily in the United States, of which 41% resulted from USDA CEAP research within three U.S. watersheds, in the Europe, within the European Union, was next most frequent with approximately 8%, while remaining global regions accounted for less than 14% of the published literature (Figure 4a). Chu et al. [38] in Canada used modeling scenarios to assess potential fish diversity responses to changes in climate and land management resulting from climate change and associated land-use changes. The study provided modeled scenarios indicating conservation management priorities, including use of CAs, would most likely increase for watersheds along the eastern and western maritime coastlines to maintain fish diversity in these regions. Chu et al. [38] concluded that, 'To be effective, conservation and management of aquatic habitats and resources should aim to keep pace with changes in the types and concentration of human activities and environmental change across the landscape.' Smiley and Gillespie [17] in their literature review on fishes in channelized agricultural headwater streams found that only 8 of 33 studies published between 1963 and 2009 provided information on the effects of CAs (dam removal, instream habitat structures, riparian buffers) on fishes in the United States, which was similar to this review's findings highlighting the need for more information on the ecological effects of CAs from Europe and other parts of the world. Globally, riparian (29.5%) and wetland (18.2%) CAs were the most frequently assessed CAs individually for aquatic ecological responses. A literature review of riparian forest CAs by Broadmeadow and Nisbet [14] assessed how riparian functions are affected by design and management of the CAs. Specifically, Broadmeadow and Nisbet [14] reviewed how riparian width and vegetative structure and species could protect aquatic ecosystems within silviculture watersheds. In general, riparian widths near 30 m provided greater protection for water

quality and greater ecosystem services such as denitrification, habitat, native animal and plant diversity, temperature moderation, and sediment removal. However, a majority of studies (31.8%) assessed a combination of CAs influencing aquatic ecological responses (Figure 4b). Few studies assessed aquatic ecological responses to more specific CAs such as buffers and tillage practices [16], water control structures [54], or integrated pest management systems [89]. Christensen et al. [37] documented that percentage agricultural land retirement through Conservation Reserve Program was positively correlated with fish community structure at smaller reach scales, but not at the watershed scale in the Minnesota River basin in Minnesota. Additionally, Christensen et al. [37] suggested their fish-habitat relationships suggested that land retirement adjacent to streams may result in improved physical habitat quality, but they did not directly assess the relationships between percentage of land retirement and physical habitat quality. The influence of CAs on physical habitat quality has not been widely assessed. Smiley et al. [86,88] found that herbaceous riparian buffers planted as part of the Conservation Reserve Enhancement Program adjacent to channelized agricultural headwater streams in Ohio only widen the riparian habitats and did not alter vegetative type, vegetative structure, or instream habitat variables. The linkages between CAs and physical habitat quality have not been widely explored because most CAs are designed to improve water quality, not physical habitat quality and as such this is an area that future research needs to explore in more detail.

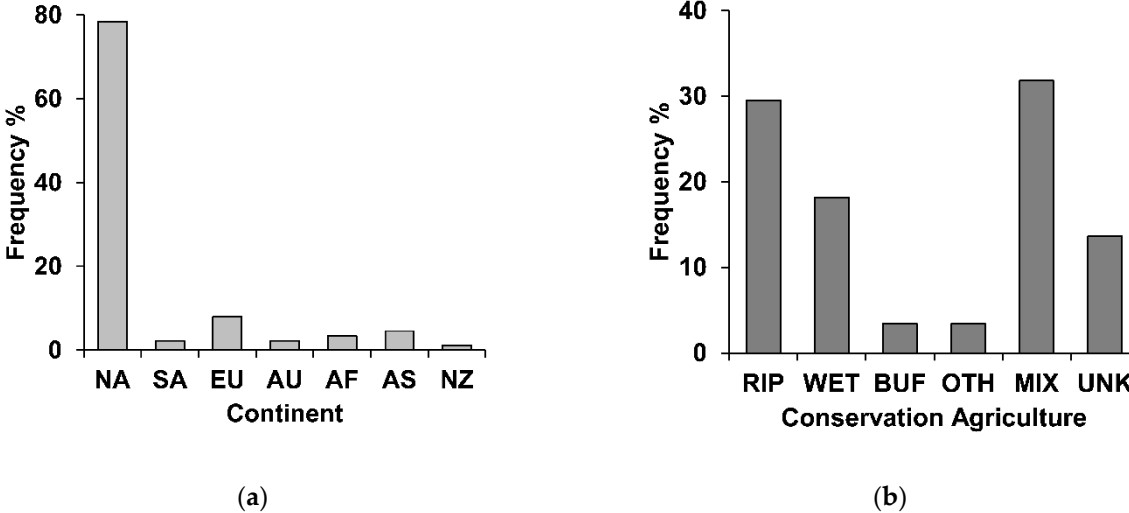

(**a**)  (**b**)

**Figure 4.** Studies from 2000–2020 assessing influence of conservation agriculture practices on aquatic ecological responses as: (**a**) Frequency across locations North America (NA), South America (SA), Europe (EU), Australia (AU), Africa (AF), Asia (AS), and New Zealand; (**b**) Frequency among conservation agriculture practices riparian (RIP), wetlands (WET), buffers (BUF), other (OTH), mixed practices (MIX), and unknown (UNK) [13–20,27–111].

The value of assessing mixed CAs is demonstrated in studied USDA CEAP stream watersheds, Cedar Creek in northeast Indiana, and Upper Big Walnut Creek in central Ohio evaluating ecological responses to conservation practices are located in the Midwestern Corn Belt, a region of highly intensive agriculture that is a significant contributor of agriculturally derived nutrients and pesticides that have resulted in the formation of hypoxic conditions in the Gulf of Mexico and the Great Lakes. These sites are also of international interest since the majority of the headwater streams in both watersheds consist of channelized agricultural headwater streams. Channelized agricultural headwater streams (i.e., agricultural drainage ditches) are common in agricultural watersheds in the Midwestern United States, Canada, and Europe [17,87,88]. These are first to third order streams that have been modified or created for agricultural drainage [17]. Channelized agricultural headwater streams are characterized by trapezoidal, straightened, widened over-enlarged channels dominated by herbaceous vegetation [17,88]. Dominant land

use in both watersheds is cropland consisting of corn (*Zea mays*) or soybean (*Glycine max*). Increased loadings of nutrients and pesticides from agricultural fields and bacteria from failed septic tanks are non-point source pollutants of concern. Notable combined (mixed) CA initiatives within these watersheds have included: (1) implementation of the USDA NRCS Environmental Quality Incentive Program (EQIP) for the improvement and/or development of animal waste systems to reduce pathogen inputs to the watershed; (2) integrated pest management for the reduction of atrazine concentrations; (3) promotion of riparian buffers via Conservation Reserve Enhancement Program to improve water quality; and (4) educational outreach initiatives to promote the adoption of agricultural best management practices [17,87,89]. Additionally, portions of Cedar Creek are designated as part of the Indiana State Natural, Scenic, and Recreational River System, which provides those reaches with protection from construction, dam, and drainage projects and has led to the upper reaches of the watershed as being designated an EQIP priority area. Another example within USDA CEAP of assessing combinations of CAs in a lake watershed is Beasley Lake in western Mississippi. Beasley Lake is a riverine lake watershed with >50% of land-use in row crop agriculture planted in soybeans, cotton (*Gossypium hirsutum*), and corn. Implemented CA practices include conservation tillage practices, specialized drainage culverts (i.e., slotted board risers) to reduce runoff flow rates, and the planting of temporally stable vegetated buffers and grassed filters, conversion of highly erodible cropland to vegetative cover via the Conservation Reserve Program, constructed wetland, wildlife habitat vegetative buffers, and a two-celled sediment retention pond (Figure 5) [63,67,70].

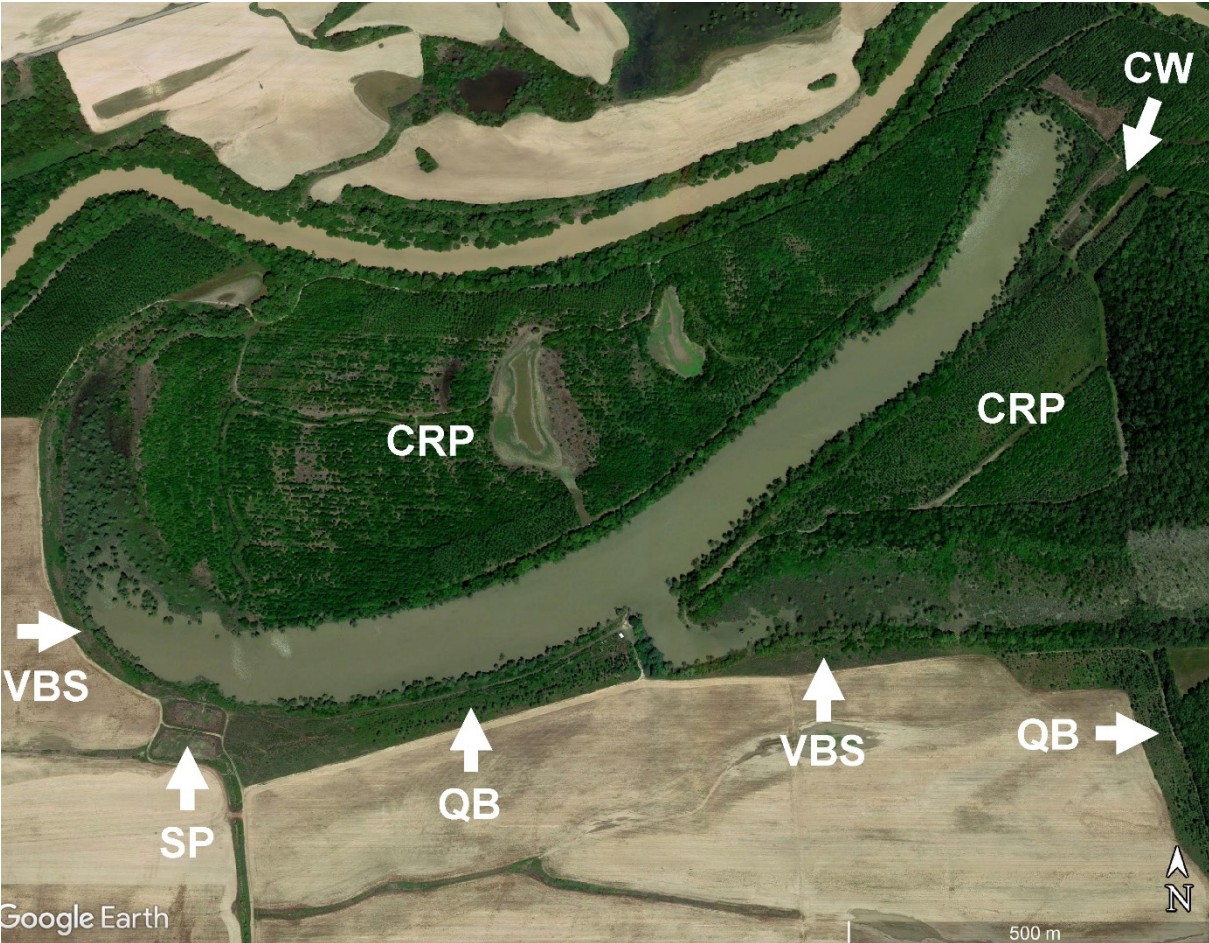

**Figure 5.** Beasley Lake Watershed in western Mississippi, USA with implemented Conservation Agriculture practices: vegetated buffers and grassed filters (VBS); conservation reserve program conversions (CRP); constructed wetland (CW); wildlife habitat vegetated buffers to attract bobwhite quail (*Colinus virginianus*) (QB); and a two-celled sediment retention pond (SP).

### 3.2. Ecological Responses to Conservation Agriculture

Synthesis of ecological responses to CAs among the 88 studies indicated that CAs can have positive, neutral, and occasionally negative influences on aquatic ecosystems. Of the studies assessed, 39.8% indicated a positive ecological response to CAs (e.g., increased biodiversity) [16,20,33,37,39,44,48,49,51,53,54,60,61,64–66,72,74–76,78,81,82,84–86,90–92,94, 99,106,111] and 40.9% had no discernable ecological influence [17,27,30,31,34,35,52,57–59,62,63,67–71,77,79,80,88,89,95–98,102,103,105,107–110]. Only two studies that evaluated fish responses [36,83] documented negative influences of CAs. For 17% of the studies assessed, ecological responses to CAs could not be determined. A summary of the common structural and functional responses used to evaluate CAs at sub-watershed and watershed scales are presented in Figure 6.

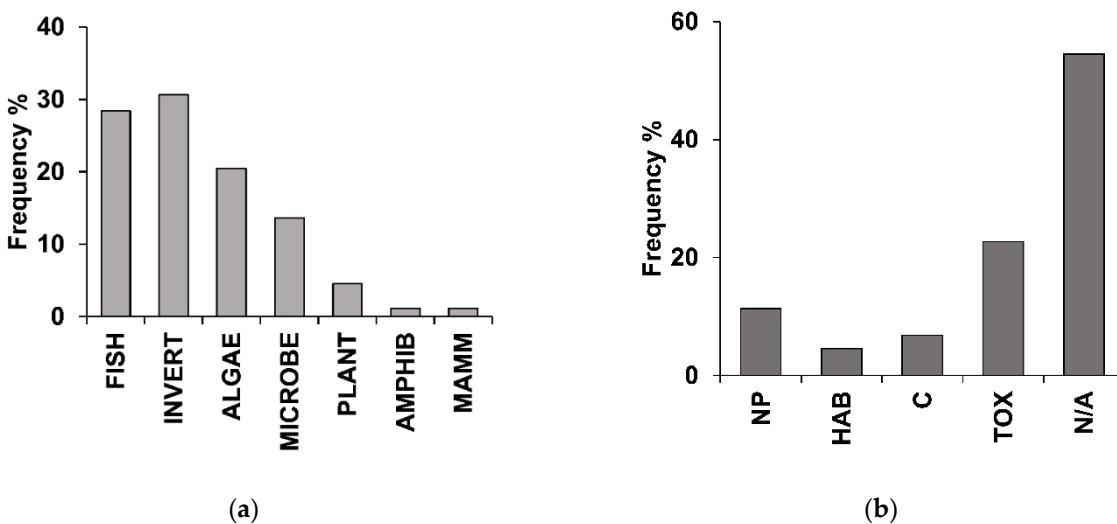

(a)   (b)

**Figure 6.** Studies from 2000–2020 assessing influence of conservation agriculture practices on aquatic ecological responses as: (**a**) Frequency of studies assessing structural ecological responses of fish (FISH), macroinvertebrate (INVERT), algae (ALGAE), heterotrophic bacteria (MICROBE), aquatic plant (PLANT), amphibian (AMPHIP), and mammal (MAMM); (**b**) Frequency of studies assessing ecological functional responses of nutrient cycling (NP), eutrophication induced harmful algal blooms (HAB), carbon cycling (C), stressor-response ecotoxicology (TOX), and non-specified (N/A) [13–19,27–111].

A conceptual model depicting linkages between CAs, influenced agricultural stressors, and potential ecological structural and functional responses is presented in Figure 7. There are a wide variety of ecological responses that can be assessed in both lotic and lentic aquatic ecosystems at a variety of spatiotemporal scales. Ecological structural responses (i.e., changes in biodiversity, abundance, or species composition) can occur at lower trophic levels of the ecosystem, such as autotrophic (algae and plants) and heterotrophic (bacteria and fungi) microorganisms, that are often the aquatic communities responding the quickest spatially and temporally to water quality changes resulting from implemented conservation practices [111]. Delineation of the literature assessing CA influences on algae accounted for about 20.5% while studies with aquatic plants were only 4.5% of the literature. Heterotrophs accounted for 13.6% and were almost exclusively bacteria. Aquatic macroinvertebrates and fish communities are sensitive indicators of ecosystem health and ecological integrity and information from these higher trophic levels of consumers can provide valuable insights regarding the impacts of the physical and chemical changes caused by conservation practices. Studies of CA effects on fish occurred in 28.4% of the studies while macroinvertebrate studies were the most frequent at 30.7%. Studies with amphibians and mammals were rare (<2%) with only one study on amphibians [88] and one study on mammals, specifically beaver (*Castor fiber*) [94], were found. Ecological functional responses include physical processes of hydrology, morphology, and physical and chemical components that affect ecosystem responses [112–118]. Aspects of biogeo-

chemistry including uptake, transport, and storage of nutrients are also important elements of lotic and lentic aquatic ecosystems that can assist with understanding ecosystem responses to agricultural conservation practices [14,49,69,112]. To date, ecological research in agricultural watersheds has encompassed a broad suite of ecological responses scaling along the hierarchy of biological organization including nitrogen biogeochemistry, microbial ecology of phytoplankton and heterotrophic bacteria, a multitude of ecotoxicological assessments ranging from standard toxicity bioassays to biomarker assessments, and fish ecology assessments involving population and community ecology (see: Figure 6a,b). However, most studies are conducted using comparisons of watersheds across a gradient of CA and/or stressor conditions, such as comparisons of ecological responses between a reference watershed and agriculturally impacted watershed. Far fewer assessments include before-after, control-impact (BACI) experimental designs [89] that provide more robust linkages between CAs and ecological responses and better forecasting of CA effectiveness. This would require pre-CA conditions being measured and, for many ecological responses, long-term (decadal) ecosystem assessments.

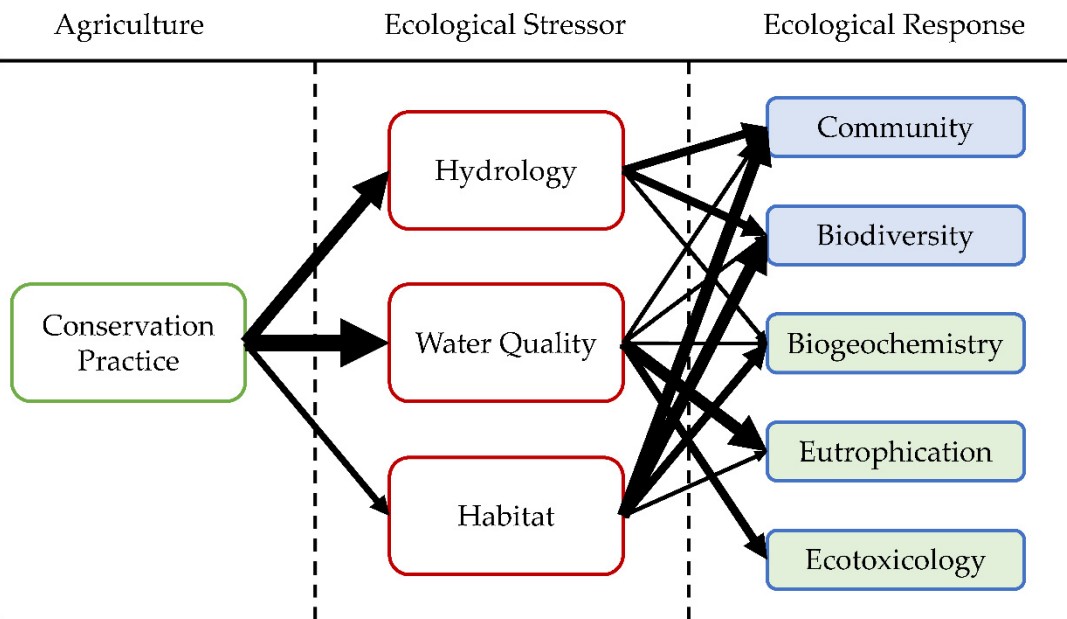

**Figure 7.** A conceptual model depicting linkages between conservation agriculture practices, or CAs (green outline box), influenced agricultural stressors (red outline boxes), and potential ecological structural (blue outline boxes with blue fill) and functional (blue outline boxes with green fill) responses. Arrow thickness indicates hypothesized strength of association between CA, ecological stressor, and ecological response.

### 3.3. Structural Ecological Responses

Structural ecological components of aquatic systems encompass the diversity, abundance, and species composition of biota from heterotrophic bacteria to algae, macroinvertebrates, and vertebrates among populations and communities. These structural components interact with the physical and chemical external aquatic environment and can be altered by agricultural implementation of conservation practices in the watershed (Figure 7) [112–115]. Macroinvertebrate and fish ecology are the two most commonly studied components in assessing environmental quality in streams and lakes [115,116] and were the most frequent ecological responses to CAs measured [16,17,29,31,36–38,40,41,43–48,51–53,55,56,58,60–62,64–66,68,70,72,74–78,81–86,89,91,92,95,108] (Figure 6a). Fish community assessments conducted in USDA CEAP watersheds indicated that implementation of single CAs (i.e., herbaceous riparian buffers or atrazine reduction practices) implemented to improve water quality conditions will not benefit fish communities within the watersheds of channelized agricultural headwater streams in the Midwestern United States [88,89].

These results are supported by extensive information on fish-habitat relationships within these streams that indicates that fish community structure is being influenced mostly by instream habitat rather than water chemistry [86,119–122]. In contrast, fishery assessments of the influence of multiple CAs (i.e., herbaceous buffers, slotted board risers, forested buffers, conservation tillage) implemented to improve water quality within an oxbow lake of the Mississippi Delta in western Mississippi, USA documented that the implementation of these practices lead to changes in fish community structure [61,62,123]. Fish communities and populations in stream ecosystems are strongly influenced by instream habitat [17,86,119,120] and other physical habitat variables (land use, watershed soil type, geomorphology, and percent fines) [121,122]. Stream channelization has resulted in physical habitat degradation that impacts fish communities within channelized agricultural headwater streams in the Midwestern United States and Canada [17]. Subsequent assessment of nutrients and pesticides with USDA CEAP watersheds in Indiana and Ohio has documented that while these streams experience periodic episodes of elevated levels of nutrient and pesticide concentrations [17,87,120,124], average and median concentrations are typically below the chronic and acute toxicity levels for fish [17,81,119]. Additionally, the implemented CAs within channelized agricultural headwater streams did not result in improved water quality conditions or physical habitat quality within these streams [88,89]. Thus, implementation of CAs designed to improve water chemistry conditions within channelized agricultural headwater streams in the Midwestern United States did not address the primary degradation factor (i.e., physical habitat degradation) influencing fishes within these streams. In contrast, the implementation of multiple CAs within USDA CEAP Beasley Lake Watershed in the Mississippi Delta of western Mississippi, USA, was specifically intended to reduce suspended sediment loads, which is the primary source of degradation within this ecosystem type [58]. The concerted implementation of multiple CAs over a 15-year period reduced total suspended sediment concentrations that led to a cascade of associated physicochemical changes such as improved visibility, increased phytoplankton productivity, and ultimately changes in the structure of fish communities and populations [61,62,123]. These combined cross-watershed results suggest implementation of CAs will not benefit fishes within small agricultural watersheds unless multiple CAs are implemented specifically to address the primary factor (or factors) contributing to the degradation of fish populations and communities. This conclusion has important ramifications within the United States for current USDA NRCS guidelines related to implementation of CAs. USDA NRCS implementation of CAs within agricultural watersheds of the United States occurs as a result of voluntary adoption of CAs by farmers and landowners. Additionally, USDA NRCS guidelines currently limit landowner compensation to the adoption of only one practice on the same property at one time. These two policies combined typically result in haphazard implementation of single CAs within agricultural watersheds in the United States that may potentially delay or hinder the recovery of aquatic ecosystems from the impacts of agriculture. USDA NRCS policy changes that enable the targeted implementation of multiple CAs within agricultural sub-watersheds are needed to positively benefit fish communities and populations.

Global climate change can have long-term and watershed-scale influences on CA effectiveness in mitigating transport of agricultural pollutants to streams and lakes. Collectively, these multiple stressors can alter the health and diversity of aquatic ecosystems and fish communities [38,45,51,82,83]. Recent research has attempted to address how expected global climate change might affect long-term effectiveness of present-day CAs to try and reduce uncertainty and protect water quality and fish biodiversity with conservation planning [39]. Recent research by Fraker et al. [45] and Hall et al. [51] in the U.S. and Canada integrated the use of several models to attempt to forecast global climate change scenarios to estimate CA effectiveness and resiliency under altered precipitation patterns relative to protecting fish community diversity. The studies indicated that changes in climate are likely to increase the need for CAs to protect diversity [51], but that efforts to improve water quality through CAs may come with costs to other ecosystem services such as unintended

shifts in fish communities [39]. Studies by Sarkar et al. [82,83] in India indicated that fish in the region were maturing earlier, due in part to warming temperatures from climate change and alterations in hydrology from human activity such as irrigation for agriculture. As a result, Sarkar et al. [83] emphasized the need to maintain and enhance riparian floodplain wetland conservation as a means to ameliorate the impacts of agriculture and climate change to protect fish diversity in watersheds of India.

### 3.4. Functional Ecological Responses

Functional aquatic ecological components incorporate ecosystem processes of biogeochemistry, eutrophication affecting nutrient cycling, food webs, and system metabolism, and cause-effect stressor-response relationships involving ecotoxicology (Figure 7) [124–127]. Approximately 45% of studies assessing ecological responses to CAs incorporated a functional ecological component [20,29,32–35,41,47,50,60,65,66,68–70,73,75–80,84,89–92,96–103,105–107,109,112]. Of these studies, four assessed harmful algal blooms [30,32,33,50], six studied components of carbon cycling [73,90,96,97,103,105], 10 examined aspects of nutrient (nitrogen, phosphorus) cycling [20,34,35,79,80,100–102,106,107,112], and the remainder assessed ecotoxicology (Figure 6b). Since most studies did not contain components of ecosystem function, this area of research is in greater need to assess how CAs could influence aquatic ecosystem processes.

Eutrophication induced by elevated nitrogen and/or phosphorus alters the function of aquatic ecosystems through a variety of pathways including, but not limited to, production of harmful algal blooms (HABs) [30,32,33,50], changes in algal nutrient limitation [34,35], hypoxia (i.e., dissolved oxygen stress induced by concentrations at or below 2 mg/L) [30,80,106], and alteration of aquatic ecosystem metabolism [79]. Ecosystem services of CAs such as wetlands and wetland vegetation allowed to grow and flourish in constructed wetlands or drainage ditches implemented in an agricultural watershed allow for the processing, uptake, mineralization, and cycling of carbon, nitrogen, and phosphorus [20,111]. Furthermore, understanding the relationships between eutrophication, nutrient source (e.g., internal catchment loading), legacy nutrients, system lag-times, and ecosystem biogeochemistry in agricultural landscapes (Figure 7) can provide greater success in CAs controlling the deleterious effects of eutrophication [79,80,100,102,106,107,112]. About half (48.7%) of the reviewed studies focused on assessments with CAs (as ecosystem services) on carbon [73,90,96,97,103,105], nutrients (N, P) [20,34,35,79,80,100–102,106,107,112], and harmful algal blooms [30,32,33,50] in relation to eutrophication. While several studies measured some CA mitigation in eutrophication [20,33,90,100,106,112], most studies had either neutral or unknown CA influences [30,32,34,35,50,73,79,80,96,97,102,103,105,107]. Nutrient dynamics in streams, rivers, and lakes of agricultural watersheds are often linked to upland terrestrial legacy nutrients such as phosphorus [6,100] and/or subsurface transport of nitrogen, often as nitrate [20,49,50]. This increase in nutrients can then alter transport and fate, increase productivity, and potentially elicit HABs [50,54]. While alteration of nutrient dynamics can occur within the lotic or lentic system [20,54,71], CAs can be utilized to harness some watershed ecosystem functions [9,20,49,107]. Denitrification and phosphorus sequestration (e.g., plant uptake) that mitigate alteration of nutrient dynamics. The framework proposed by Goeller et al. [49] using multiple stacked CAs combined with in-stream management to enhance ecological processes such as denitrification could be used to control nutrients such as nitrogen. Also, managing channelized agricultural headwater streams for both drainage (i.e., drainage ditches) and enhanced ecological processes can be enlisted to further control excess nutrients that alter nutrient dynamics in streams, rivers, and lakes [20]. Currently, a research gap remains between measured reductions in nutrients with CAs and how these reductions shift or alter nutrient dynamics in impacted agricultural watershed water bodies and future research is needed.

Biogeochemical processes can be harnessed within a framework of integrated or stacked CAs and stream habitat restoration to improve nutrient cycling and increase nutrient attenuation in agricultural watersheds. Goeller et al. [49] proposed a framework

of stacked CAs including edge-of-field practices (e.g., grassed filter strips, retention ponds), riparian buffers (e.g., constructed wetlands) combined with in-stream habitat structures (e.g., woody debris, low-grade weirs) to enhance biogeochemical attenuation of nitrogen in New Zealand agricultural watersheds. Goeller et al.'s [49] framework is flexible to allow researchers, stakeholders, land-owners, and farmers a variety of options that are field to farm-specific and increase the likelihood for success in assessing the most effective measures for mitigating farm nutrient runoff. Biogeochemistry (e.g., nutrient cycling) studies conducted in USDA CEAP watersheds included Ullah et al. [99] 2005 comparison of denitrification rates and $N_2O$ emission rates (greenhouse gas) in cultivated row-crop soils to mature bottomland hardwood riparian forest soils within Beasley Lake Watershed in western Mississippi, USA. With the addition of $NO_3$, riparian forest soils showed a more than two-fold increase in denitrification potential compared with cultivated soils. In contrast, $N_2O{:}N_2$ emission ratios were consistently greater in cultivated soils than in riparian forest soils with or without the addition of $NO_3$. These results indicated that differences in land use leads to differences in the watershed's capacity for denitrification and is a direct result of different physiochemical properties of the two soil types where wetter soils, greater soil porosity, and greater soil organic carbon loads found in forested riparian soils significantly increases denitrification compared with cultivated soils. In addition, many of these same soil properties also allow greater N mineralization and lower levels of $N_2O$ emission in forested riparian soils further highlighting the value of using forested riparian buffers as a CA practice. Similarly, Keating et al. [100] measured soil denitrification gene (*nosZ*) abundances and $N_2O$ emissions in CAs and cropland within Beasley Lake Watershed in Mississippi during summer 2013. Keating et al. [100] observed highest $N_2O$ in cropland and riparia, although measured soil genetic and chemical data suggested a difference in $N_2O$ sources between these two sites. Denitrification gene (*nosZ*) abundances in soils indicated denitrification-derived $N_2O$ products in intermittently flooded wetland and riparia sites, while soil inorganic nitrogen concentrations indicated greater nitrification-derived $N_2O$ products in dry cropland sites in the watershed. A more comprehensive assessment of soil denitrification potential across land-use types was conducted in a USDA CEAP Beasley Lale Watershed from 2002 to 2003 by Ullah and Faulkner [101]. Multiple land-use types including upland cultivated soils, vegetated and unvegetated channelized headwater stream soils, constructed wetland soils, and forested riparian soils were examined. Soil denitrification potentials were 6.3-fold greater in forested riparian soils and two-fold greater in constructed wetland soils than comparable cultivated soils [101]. Additionally, denitrification potentials within soils collected from vegetated and unvegetated channelized headwater streams were less than wetland and forested riparian soils [101]. These results provide support for the use of forested riparian soils and constructed wetlands to mineralize nitrogen and reduce nitrogen runoff into agricultural lakes like Beasley Lake.

Examples of heterotroph functional ecology was observed with Zablotowicz et al. [108] that documented that a significant fraction (33–100%) of Beasley Lake water borne populations of the colony-forming heterotrophic gram-negative bacterioplankton, fluorescent *Pseudomonas* sp., were capable of metabolizing and/or co-metabolizing three commonly used herbicides, metolachlor, propanil, and trifluralin. Metolachlor was transformed by fluorescent *Pseudomonas* sp. via glutathione conjugation, while propanil was transformed via aryl acylamidase hydrolosis and trifluralin was transformed via aromatic nitroreduction. Although herbicide biotransforming fluorescent *Pseudomonas* sp. did not exhibit responses to specific CAs within Beasley Lake, these results are important because they indicate that CAs that can result in increased abundance of fluorescent *Pseudomonas* sp. may help with decreasing herbicide concentrations within aquatic ecosystems. One of the more detailed microbial studies conducted in Beasley Lake included assessments of extra cellular enzyme activity such as fluorescein diacetate hydrolysis (1997 to 1999), alkaline phosphatase, and substrate utilization (2000 to 2003) [109,128], as well as evaluations of phytoplankton and bacterioplankton populations with a suite of diagnostic photosynthetic

pigment biomarkers (e.g., chlorophylls, xanthophylls, carotenoids, etc.) and spiral plating assays (2000 to 2003). Despite the extensive microbiological assessment, observations of heterotrophic bacteria and associated extra cellular enzyme activity were similar to those of Cullum et al. [39] where implementation of CAs from 1995 to 2003 appeared to have minimal impact on these microbial endpoints [109,128]. Cullum et al. [39] did not observe any effects of CAs on either coliform or enterococci bacteria counts. However, algal biomass, as chlorophyll concentration, clearly increased four years after implementation of CAs coinciding with decreases in suspended sediments indicating algal populations were primarily light limited but not nutrient limited [39].

Functional ecological response studies focusing on ecotoxicology accounted for 51.3% of these studies and exclusively examined the effects of CAs on mitigating pesticide effects both in surface waters and river or lakebed sediments (Figures 6b and 7). Ecotoxicology research was conducted primarily (70%) on macroinvertebrate responses (communities to individuals) [29,47,60,65,66,68,76–78,84,89,91,92,98] with only 15% of studies assessing fish and 15% assessing microbial (algae and/or bacteria) responses [41,70,89,98,109]. Studies by Gagliardi and Pettigrove [47] in Australia, Schäfer et al. [84] in Europe, and Tsaboula et al. [98] in Greece, were among the more comprehensive ecosystem assessments of pesticides on non-target aquatic organism monitoring and assessing suites of current-use and/or legacy pesticides (e.g., ΣDDT) at dozens to >100 river and stream sites to demonstrate pesticide mitigation using CAs. Additionally, a number of studies conducted at various scales in the United States attempted to directly link more specific CAs with pesticide mitigation and ecological impacts.

Ecotoxicological bioassays and biomarker assessments conducted within channelized agricultural headwater streams in the Midwestern United States [41,81,119] and oxbow lakes of the Mississippi Delta in western Mississippi, USA [60,65,66,75–77] provided indications of the potential for sublethal effects of agricultural contaminants on the biota within small agricultural streams and lakes. Experimental animals in these bioassays and biomarker assessments were exposed to realistic concentrations of agricultural contaminants as a result of being exposed to water or sediment obtained directly from USDA CEAP sampling sites. This is a significant contrast to typical laboratory assessments of agricultural contaminants with experimental animals exposed to a wide range of concentrations that do not represent real-world conditions. A common finding among all the USDA CEAP ecotoxicology research in both watersheds was that exposure to realistic concentrations of agricultural contaminants below acute and chronic toxicity levels did not reduce survival of experimental animals, but instead resulted in the occurrence of sublethal responses involving reduced growth or reproduction. The potential for sublethal effects within aquatic macroinvertebrates and fishes suggests that CAs that can lead to water quality improvements below the current regulatory standards may benefit the biota within small agricultural streams and lakes. From a management perspective, this cross-watershed conclusion suggests that CA plans for small agricultural watersheds that attempt to go beyond simply meeting the state and federal regulatory requirements for water quality will provide greater ecological benefits than those that simply attempt to meet the regulatory requirements. From a research perspective, this cross-watershed conclusion suggests that ecological responses to water quality improvements as a result of CAs are likely to be expressed in more subtle ways than ecological responses to extreme physical habitat degradation or chemical contamination. Sublethal responses to water chemistry changes are difficult to detect [117] and future assessments of CAs will require concerted research efforts involving tiered assessments across watershed, field, mesocosm, and laboratory scale experiments.

## 4. Conclusions

Our summaries and synthesis of CAs and aquatic ecology research results across watersheds and ecosystem types provided insights regarding the ecological effects of CAs that can be used to reduce the potential impacts of agriculture on streams, rivers, and lakes.

Key findings from this synthesis highlight the importance of using combinations of CAs to address primary factors responsible for ecosystem degradation and the importance of tiered assessments to detect subtle, sublethal responses of the biota to water chemistry changes as a result of conservation practices.

Despite the body of information produced by ecological research effort over the past 20 years, there are still several critical information gaps of ecological processes and functions that still need to be addressed to improve our understanding of the ecological effects of CAs at the watershed scale. Notable gaps include: (1) improved understanding of how stream and lake nutrient spiraling, cycling, and internal loading (retention) are affected by CAs; (2) clarifying how and to what extent stream and lake trophic states and interactions are altered with changes in water quality; (3) better understanding of the effect of CAs on physical habitat quality as well as water quality; (4) the influence of changing nutrient-pesticide mixtures on the biota as a result of water quality improvements due to implementation of CAs; (5) documentation of which combinations of CAs provide the greatest ecological benefits; and (6) quantifying the minimum percentages of agricultural watersheds that need to receive implementation of CAs to mitigate the effects of agriculture on the biota within agricultural watersheds. Particularly, new research efforts involving the individual (e.g., genetic, biochemical, physiological) and population level (e.g., reproductive output) impacts of agricultural contaminants would assist with understanding the sublethal impacts of agricultural contaminants and the potential benefits of water quality improvements beyond the current regulatory standards. Novel research that documents the influence of CAs on ecosystem structure and function within and among watersheds would be useful to further our understanding of whether conservation practices can restore aquatic ecosystems. One example would be expanding existing research tools and techniques to assess nutrient dynamics from multiple stacked or integrated CAs to runoff to downstream receiving waterbodies using geographic information system framework with empirically measured denitrification rates across the landscape and into streams, rivers and lakes to map hot spots of nitrogen transport and transformation. Coinciding with this would be assessments of HABs nutrient limitation and nutrient sensitivies to determine how nitrogen affects HABs and possible nitrogen thresholds needed to mitigate HABs. Another example would be to expand the above efforts to include carbon cycling and aquatic system metabolism using state-of-the-art dissolved oxygen monitoring and models across an empirically measured (e.g., area in ha) gradient of CAs within or among multiple watersheds. A third example would be utilizing and expanding research networks across disciplines (engineering, biology, soil science, agronomy, social science) and multiple watersheds at national and international scales. For example, USDA-ARS Long-Term Agroecosystem Research Network (LTAR; https://ltar.ars.usda.gov/ (accessed on 9 June 2021)) can provide research opportunities to examine the influence of CAs on ecosystem services at large (continental) scales and include scientists from government, academic, and private sectors.

Although water samples for measurement of nutrients and pesticides are collected across a variety of watersheds globally, there is currently a lack of coordinated effort across watersheds to assess similar and comparable ecological endpoints. Development of more coordinated ecological assessments globally across watersheds would strengthen research efforts and other similar research initiatives designed to evaluate the influence of CAs and restoration practices. Guidelines for developing ecological assessments of CAs are in place [19] and available to assist with developing new coordinated ecological assessments. Ideal ecological endpoints are those that are ecologically relevant, accurate, precise, repeatable, and can be applied and/or implemented widely across multiple watersheds simultaneously. Too often, ecological research on CAs is site or watershed specific with ecological assessments focusing on localized impairment of water quality, habitat, or other ecosystem services. Although such studies are important for determining ecological impacts within a specific site or watershed, results are often limited and do not readily translate across multiple watersheds.

**Author Contributions:** R.E.L. and P.C.S. conceptualized this review paper. R.E.L., P.C.S., R.B.G., and S.S.K. were involved with writing, reviewing, and editing the manuscript. All authors have read and agreed to the published version of the manuscript.

**Funding:** This research was funded in part by the USDA NRCS Conservation Effects Assessment Project.

**Institutional Review Board Statement:** Not applicable.

**Informed Consent Statement:** Not applicable.

**Data Availability Statement:** No new data were created or analyzed in this study. Data sharing is not applicable to this article.

**Acknowledgments:** Michael Jenkins, Mark Jordan, and Mark Williams reviewed an earlier draft of this manuscript and provided helpful comments. Current and past personnel from the USDA-ARS National Sedimentation Laboratory, USDA-ARS Soil Drainage Research Unit, Indiana University Purdue University Fort Wayne, and USDA-ARS National Erosion Research Laboratory assisted with field and laboratory work for Beasley Lake, Upper Big Walnut Creek and Cedar Creek water chemistry and ecology sampling. Particularly, we thank Trina Harkenrider, Kathyrn Sanders, Daragh Deegan, Deepal Patel, and Mark Jordan for their analyses of the Cedar Creek bioassay and biomarker study data as well as Sam Testa and Terry Welch for the numerous ways in which they aided with Beasley Lake data collection. Landowner, site, and watershed information were provided by Natural Resources Conservation Service and Soil and Water Conservation Districts in Delaware County (Ohio), Morrow County (Ohio), Allen County (Indiana), Dekalb County (Indiana) and Noble County (Indiana). The Department of Biology at Indiana University-Purdue University Fort Wayne provided partial funding for the CC ecology research. We also are grateful to those landowners who gave permission to work on their property. Disclaimer: The use of trade, firm, or corporation names is solely for the information and convenience of the reader. Mention of names does not constitute an official endorsement or approval by the USDA or the Agricultural Research Service of any product or service to the exclusion of others that may be suitable. The USDA prohibits any discrimination in all its programs and activities on the basis of race, color, national origin, age, disability, sex, marital status, familial status, parental status, religion, sexual orientation, genetic information, political beliefs, reprisal, or because any part of an individual's income is derived from any public assistance program.

**Conflicts of Interest:** The authors declare no conflict of interest.

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
