# Peer review of "Agricultural Conservation Practices and Aquatic Ecological Responses"

_water, doi:10.3390/w13121687_

Round 1
Reviewer 1 Report
See attached comments to authors.

Author Response
WATER – 1247233
COMMENTS REVIEWER [1]
Suggestions to Authors:
- There seems a missed opportunity within the introduction to familiarize those interested in agricultural intensive system management with terms that distinguish “healthy” within systems and more particularly, ecological effects and their outcomes on ecosystem-scale assessments. The literature seems to be overlooked early as to the current state of ecosystem management as it might pertain to watershed scale issues or the importance of forest management and inclusion as mitigation systems. Some beginning references in this regard might include: Peter Chapman, 1992 (Integrative Assessments in Aquatic Ecosystems), Penaluna et al., 2018 and Shively et al., 2018 (Forest impacts on watershed management), Schmutz and Sendzimir, 2018 (Aquatic ecological responses), Costanza and Mageau, 1999 (What is a healthy ecosystem?), Larned and Schallenberg, 2019 (stressor response variables related to management of aquatic ecosystems). Ecological effects, ecosystem responses and healthy aquatic ecosystems seemed muttled without links that bring the reader into a basic understanding of the gaps and data needs- that are well covered in the review.
Lines 45-67 revised manuscript: As suggested in the first paragraph of the Introduction we added text and supporting citations that define healthy watersheds for the readers and how typical management of agricultural watersheds differs from that of public wilderness tracts. Additionally, as suggested we used some of the suggested citations (Shively et al. 2018, Schmutz and Sedzimir 2018, Costanza and Mageau 1999,) above in support of the new content. Additionally, as suggested we revised the second paragraph and provided the readers an overview of what we mean by ecological effects and how it is documented. Additionally, in this second paragraph we used Larned and Schallengberg (2019) as a supporting citation. Thus, our revisions above clarify healthy aquatic ecosystems, ecological effects and ecosystem responses for the readers.
Also see revised References section, lines 650-676
- The above considerations will then handle the redundancy that currently exists in the review between “quantitative links between varying CAs and ecological responses” and “linkages of CAs with aquatic ecosystem structure and function”.
Lines 45-67 revised manuscript: We concur that our revisions to paragraph 1 and 2 of the Introduction as described above will address the reviewer’s concern here.
- Line 53, Consider The USDA initiated the …..(CEAP) as a multi-agency collaboration…
Line 120 revised manuscript: We revised this sentence as suggested.
- Line 82, …student graduate theses and…
Line 149 revised manuscript: We revised this sentence as suggested.
- Line 98, Confusing order of item with “c) denitrification”, especially since already listed.
Lines 166-167 revised manuscript: As noted in our response to Reviewer 1 comments – we have corrected this sentence by deleting “and c) denitrification” and inserting “and” before “g) denitrification.
- Line 120, Consider …studies were conducted in North America and primarily in the United States, of which…
Lines 189-190 revised manuscript: We revised this sentence as suggested.
- Line 122, Space needed after sentence completion with U.S. Currently runs into Europe, within…
Line 191 revised manuscript: As suggested we revised the end of this sentence, but made an additional change to make it more concise so it now reads “…USDA CEAP research within three watersheds. Europe, ….
- Line 128, …which was similar to this review’s findings, highlighting the …
Lines 205-206 revised manuscript: We revised this sentence as suggested.
- Line 130, …the most frequently assessed (what?) individually for… (As currently stated, seems as though the term is missing for a parameter that was individually assessed?)
Line 208 revised manuscript: we included the term ‘CA’ as this is the parameter that was assessed.
- Line 158, Consider Additionally, portions of Cedar Creek are designated as part of the Indiana State…System, which provides those reaches….
Lines 260-261 revised manuscript: We revised this sentence as suggested.
- Line 161, …watershed being designated (drop the “as”)
Line 264 revised manuscript: We revised this sentence as suggested.
- Line 166, out of place space after …flow rates,…
Line 269 revised manuscript: As suggested we have deleted the extra unneeded spaces in this line
- Line 177, drop the a before “fish” and the comma after the [30,77].
Line 2283 revised manuscript: We revised this sentence as suggested.
- Line 190, Is it “biodiversity shifts” that is being referred to here? Or other structural reponses? Also, a comma needed after …ecosystem, such…
Lines 295-297 revised manuscript: We have revised this sentence to clarify what we mean by ecological structural responses and added the needed comma after ecosystems. We also revised this sentence so that we only have one “such as” within the sentence and as such this will help with clarifying our main thought. The sentence now reads as follows: …structural responses (i.e., changes in biodiversity, abundance, or species composition) can occur at lower trophic levels of the ecosystem, such as…..
- Line 191-192, Consider, …microorgansisms often being the aquatic communities responding significantly both spatially and temporally to water….
Lines 297-300 revised manuscript: We concur with most of the rewording suggestion here, but not the replacement of “quickest” with “significantly”. For us “significantly” is problematic because it has specific statistical connotations and it is vague. The supporting citation we reference here indicates that microorganisms respond the quickest and as such we chose to keep that descriptor, but have incorporated all other editorial suggestions here. The sentence now reads ….. autotrophic (algae and plants) and heterotrophic (bacteria and fungi) microorganisms, that are often the aquatic communities responding the quickest spatially and temporally to water quality….
- Line 219, comma needed following “conditions, such as…agriculturally-impacted watershed…
Lines 327-331 revised manuscript: As suggested we have added the comma after conditions and have made some minor revisions here to more clearly describe the experimental design and the sentence now reads …conditions, such as comparisons of ecological responses between reference watershed and agriculturally impacted watershed.
- Line 220, Far fewer assessments include before-after, control-impact….that provide more…
Lines 329-331 revised manuscript: We revised this sentence as suggested.
- Line 223, …conditions being measured…, long-term (decadel) ecosystem assessments.
Line 333 revised manuscript: We revised this sentence as suggested.
- Line 228, vertebrates (spelling)
Line 338 revised manuscript: As suggested we corrected the spelling of vertebrates
- Line 240, …indicates fish community structure being influenced…
Line 350 revised manuscript: We revised this sentence as suggested.
- Line 248, …morphology and percent fines)
Line 358 revised manuscript: We revised this sentence as suggested.
- Lines 278-280, Authors might consider some of the ongoing literature reporting critical policy changes that target conservation efforts within watersheds (even international shifts). The literature is extensive here on water-related ecosystem services (Guo et al., 2021; Tolessa et al., 2017; Leh et al. , 2013; Torres et al., 2021).
While we greatly appreciate the reviewer highlighting the importance of policy changes affecting conservation efforts, this is a topic that is specifically political and beyond the scope of the review paper which is focused on the scientific reporting of conservation practices and ecological responses. Because of this, we made no additional changes to the manuscript.
- Line 306, Line word spacing issue as shown
Line 416 revised manuscript: We checked this line and could not find any extra spaces to remove, thus the appearance of extra spaces is result of using the justified page alignment, which follow’s Water’s formatting requirements. No changes made.
- Line 312, Run on sentence. (that, which, ?) …compared
Lines 465-468 revised manuscript: As suggested we have revised this sentence so it now reads …..included Ullah et al. [93] 2002 comparison of denitrification rates and N2O emission rates (green house gas) in cultivated row crop soils to mature bottomland hardwood riparian forest soils within Beasley Lake watershed in western Mississippi, USA.
- Line 318, Consider, These results indicate that differences…
Lines 471-472 revised manuscript: We revised this sentence as suggested.
- Line 337, As with item 24, run on with - …that documented…
Line 497 revised manuscript: As suggested we inserted “that” between [101] and “documented”
- Line 347, Vague use of “broader” study involving microbial studies.
Line 508 revised manuscript: As suggested we have replaced “broader” with “more detailed”
- Line 349, comma needed after [102,121], as well as…
Line 516 revised manuscript: We revised this sentence as suggested.
- Line 364, …organism monitoring…
Line 529 revised manuscript: We revised this sentence as suggested.
- Line 366, …sites to demonstrate pesticide…
Line 531 revised manuscript: We revised this sentence as suggested.
- Line 367, …conducted at various scales in the United States attempted to directly link…
Lines 532-533 revised manuscript: We revised this sentence as suggested.
- Line 378, …contaminants with experimental animals exposed to…represent realistic conditions.
Line 543 revised manuscript: We revised this sentence as suggested.
References section had several lines with word spacing issues that may be a fragment carry-over of programs or need attention for final editing. Those lines included: 476, 482, 512, 569, 657, 664, 667,
We checked these lines and other places in the References section appearing to have word spacing issues and could not find any extra spaces to remove, thus the appearance of extra spaces is result of using the justified page alignment, which follow’s Water’s formatting requirements. No changes made.

Reviewer 2 Report
Dear Authors.
This is a very interesting review, well written, and nicely presented.
No comments on my part for this interesting review work.
Author Response
WATER – 1247233
COMMENTS REVIEWER [2]
Comments and Suggestions for Authors
Dear Authors.
This is a very interesting review, well written, and nicely presented.
No comments on my part for this interesting review work.
The authors sincerely thank the reviewer for volunteering to assess the manuscript and help the authors improve the quality of the publication. No changes made.

Reviewer 3 Report
This study focus on a review analysis of conservation agriculture practices. The study is interesting, however, requires the following revisions.
Abstract: Please give some data related to your research/findings.
The introduction is short. Improve the content.
Line 96-99……Revise…….Check “above CA variables”……Use of comma after further…..Check “g) denitrification; and c) denitrification.”
Line 121-122…..The authors can remove full stop in Midwestern U.S. and Southeastern U.S.
Line 149…….Write the scientific names in italics.
What is the source of Figure 1?
The Materials and Methods section may be changed to Methods only.
Elaborate on the number of studies considered for the study as per the year (at least highest and lowest) in Methods section.
Only two databases were considered??
It would be better to also mention the conservation practices taken by the researchers. That would also demarcate the practices that are more suitable in maintaining water quality.
The study claims that most relevant studies were conducted in USA. Was there any difference between USA and other countries related to water quality indicators?
Line 311…..Give comma after e.g.
The authors may present their findings in a table on “what CAs are being assessed, and what ecological responses are being measured” for easy understandings.
A conclusive statement related to Ecological function keywords like eutrophication, denitrification, etc. is missing.
Author Response
WATER – 1247233
COMMENTS REVIEWER [3]
Comments and Suggestions for Authors
This study focus on a review analysis of conservation agriculture practices. The study is interesting, however, requires the following revisions.
Abstract: Please give some data related to your research/findings.
Lines 26-41 revised manuscript: The authors note that the journal Water only allows abstract to contain 200 words or less. The abstract was re-written to incorporate the suggested additional data related to the findings of the review paper in the abstract but also not exceed the 200-word total as required for the journal Water as follows.
‘…Conservation agriculture practices (CAs) have been internationally promoted and used for decades to enhance soil health and mitigate soil loss. An additional benefit of CAs has been mitigation of agricultural runoff impacts on aquatic ecosystems. Countries across the globe have agricultural agencies that provide programs for farmers to implement a variety of CAs. Increasingly there is a need to demonstrate that CAs can provide ecological improvements in aquatic ecosystems. Growing global concerns of lost habitat, biodiversity, and ecosystem services, increased eutrophication and associated harmful algal blooms are expected to intensify with increasing global populations and changing climate. We conducted a literature review identifying 88 studies linking CAs to aquatic ecological responses since 2000. Most studies were conducted in North America (78%), primarily the United States (73%), within the framework of the USDA Conservation Effects Assessment Project. Identified studies most frequently documented macroinvertebrate (31%), fish (28%), and algal (20%) responses to riparian (29%), wetland (18%), or combinations (32%) of CAs and/or responses to eutrophication (27%) and pesticide contamination (23%). Notable research gaps include better understanding of biogeochemistry with CAs, quantitative links between varying CAs and ecological responses, and linkages of CAs with aquatic ecosystem structure and function…’
The introduction is short. Improve the content.
The authors have provided more detailed information for the following paragraphs, including added information for some references, to improve the introduction content as follows.
Lines 46-83 revised manuscript: ‘…Healthy aquatic ecosystems are sustainable ecosystems that exhibit resilience in their structure (i.e., biodiversity) and function (i.e., organic matter processing) in response to external stress [1] and subsequently able to provide a variety of ecosystem services including clean water, climate regulation, habitat for plants and animals for wildlife, carbon sequestration, nutrient cycling, and productivity [2-4]. Ecosystem. management approaches that focus on maximizing one ecosystem service result in declines of biodiversity and other ecosystem services [5] Agricultural land use within agricultural watersheds has impacted the ecosystem structure and function of lentic and lotic ecosystems via altered hydrology and increased erosion as a result of land use change and channelization and increased pollution resulting from excess inputs of nutrients, pesticides, and other agricultural contaminants [6]. Subsequently, there is an interest in reducing the impacts of agriculture on aquatic ecosystem structure and function with the use of conservation agriculture practices (CAs) that have been widely implemented in developed regions of the world including the Europe, North America, Asia, and Australia [7-10] through various agricultural agency conservation programs. However, developing countries often have very low CA implementation that greatly limits effective water resource management [11]. Additionally, most agricultural land within the developed regions of the world are privately owned and subsequently the approach towards managing agricultural watersheds differs from that of public wilderness tracts. The management of agricultural watersheds focuses on cropland pro-duction and addressing the subsequent water quality issues via the voluntary adoption by individual landowners of CAs and management of public wilderness tracts focuses on conservation via management by government agencies [3, 12]…’
The authors have provided better description of McPhee et al. [15] by including the following in the Introduction.
Line 90 revised manuscript: ‘…The French Ministry of Agriculture and Food, Territoires d’Innovation projects are agroecosystem living laboratory approaches to enhance innovation in sustainability and resilience to protect soil and biodiversity in agricultural watersheds [15]. Similarly, AAFC also utilizes an agroecosystem living laboratory approach to address agriculturally sourced environmental issues affecting soil and water management and biodiversity with a changing climate. The AAFC living labs projects allow rapid adoption of sustainable practices through close collaboration among researchers, stakeholders and land-use managers (e.g., farmers) [15]…’
The authors have provided better description of Jiao et al. [16] by including the following in the Introduction.
Line 97 revised manuscript: ‘…Within China, Chinese Nationally Important Agricultural Heritage Systems (China – NIAHS) are selected agricultural systems that demonstrate long-term (at least 100 years) sustainable historic agricultural practices providing resilience to extreme conditions (e.g., drought, flooding). These valuable heritage systems provide valuable lessons for biodiversity conservation, soil and water conservation, climate regulation and nutrient cycling while providing food and livelihood security for the rural community [16]…’
The authors have provided better description of Townsend et al. [17] by including the following in the Introduction.
Line 103 revised manuscript: ‘… In Australia, Townsend et al. [17] assessed economic valuation of multiple ecosystem services through tradeoffs between reforestation and agricultural land use as payments for ecosystem services (PES) in a process called ‘bundling’. Ecosystem services via reforestation could include water conservation, carbon sequestration, eco-tourism and conservation of biodiversity. Townsend et al. [17] suggested that the greatest likelihood for success of such a program would need to be through government establishing appropriate mechanisms to subsidize PES payments for water quality improvement.
The authors have provided better description of McDowell et al. [18] by including the following in the Introduction.
Line 110 revised manuscript: ‘…McDowell et al. [18] discussed in detail New Zealand’s water quality policy as outlined in the country’s National Policy Statement on Freshwater Management that requires integrated and sustainable water resource management. New Zealand has a combination of mandatory regulation and voluntary initiatives coinciding with monitoring and evaluation assessment programs to demonstrate successful implementation and regulatory compliance. By comparison, the United Kingdom (UK) has many similar policies and programs as New Zealand. However, unlike New Zealand, the UK utilizes subsidy payments or financial incentives in addition to the previous policies and programs to further encourage stakeholders and land managers to investment in conservation agriculture practices…’
Line 96-99……Revise…….Check “above CA variables”……Use of comma after further…..Check “g) denitrification; and c) denitrification.”
Line 164-167 revised manuscript: The authors have revised the statement from ‘…with CA variables above…’ to ‘…with above CA variables…’ as suggested by the reviewer. The authors have reviewed the syntax and grammar of the sentence and conclude that there is no need to insert a comma after ‘further’. No changes made. The authors agree with the reviewer that denitrification is included twice in the same sentences as both ‘g)’ and ‘c)’ at the end of the sentence. The reference to ‘c) denitrification’ has been deleted and the revised as follows.
From ‘…g) denitrification; and c) denitrification…’ to ‘…and g) denitrification…’
Line 121-122…..The authors can remove full stop in Midwestern U.S. and Southeastern U.S.
Line 191-192 revised manuscript: The authors agree to focus the comparison only with U.S. and other global regions and not focus explicitly on regional locations within the U.S. The sentence was revised as follows. From ‘…within two watersheds in the Midwestern U.S. and one watershed in the Southeastern U.S…’ to ‘…within three watersheds in the U.S…’
Line 149…….Write the scientific names in italics.
Lines 250, 266, 274, 309 revised manuscript: The authors have italicized the scientific name for these organisms.
What is the source of Figure 1?
Lines 131-132 revised manuscript: Figure 1. The authors note that the source of the map is from a U.S. Department of Agriculture, Natural Resources Conservation Service agency website with the following URL:
https://www.nrcs.usda.gov/wps/portal/nrcs/main/national/technical/nra/ceap/ws/
This information has now been included in the Figure 1. legend description below the figure. The map is open access and available to the public and does not require special permission to publish.
The Materials and Methods section may be changed to Methods only.
Line 146 revised manuscript: The authors agree and have changed the section title to ‘Methods’ only as suggested by the reviewer.
Elaborate on the number of studies considered for the study as per the year (at least highest and lowest) in Methods section.
Lines 171-172 revised manuscript: The authors have included additional information on the highest and lowest number of studies considered per year as suggested by the reviewer as follows. ‘…No published studies were observed for the years 2000 and 2018 while the most published studies occurred for the years 2008 and 2012 with 10 and 13 publications, respectively…’
Only two databases were considered??
Lines 151-152 revised manuscript: The authors are aware that there are more than two publication database search engines available. However, some publication database search engines are linked with the two databases used, for example BioOne and the USDA National Agricultural Library (NAL). The authors also used both aforementioned publication database search engines for the current study but did not list these since they both were linked with the two primary publication database search engines used (Google Scholar and SCOPUS). For clarification, the authors have revised the manuscript to include BioOne and NAL as additional publication database search engines used in the Methods section. Other publication database search engines that were available but not used include CORE, that searches only open access research papers and is not broader and more inclusive, and DeepDyve which is a subscription only publication database search engine.
As a result, the authors have revised the ‘Methods’ section detailing the publication database search engines as follows. From ‘…These were obtained through web-based search engines including SCOPUS and Google Scholar…’ to ‘…These were obtained through web-based search engines including SCOPUS, Google Scholar, BioOne, and the U.S. Department of Agriculture National Agricultural Library (NAL)…’
It would be better to also mention the conservation practices taken by the researchers. That would also demarcate the practices that are more suitable in maintaining water quality.
The authors note that this information is clearly reported and discussed in greater detail in the Results and Discussion section of the manuscript along with the reported most frequently assessed conservation practices in the published literature as shown in Figure 4b. For these reasons, the authors have made no additional changes to the manuscript.
The study claims that most relevant studies were conducted in USA. Was there any difference between USA and other countries related to water quality indicators?
The authors note that there were not many differences between the fewer studies in other countries and those conducted within the USA. This indicates that many of the principles of conservation agriculture and biological response paradigms, such as controlling nutrients to control eutrophication, are applicable across broad landscapes. Where the differences do occur are with more local specific economic, climatic, geographical, and biological responses (resulting from genetic diversity). Other differences that might occur have more to do with lack of resources to conduct broader or more in-depth studies in some countries.
Line 311…..Give comma after e.g.
Line 463 revised manuscript: The authors have inserted a comma after ‘e.g.’ as follows ‘e.g.,’ as suggested by the reviewer.
The authors may present their findings in a table on “what CAs are being assessed, and what ecological responses are being measured” for easy understandings.
The authors note that this would seem redundant as this information is provided both in the text (body) of the manuscript as well as in several figures including, but not limited to: Figure 4b (lines 233-236 revised manuscript), new added Figure 5 (lines 272-274 revised manuscript) and revised figures 6a and 6b (lines 271-275 revised manuscript). For these reasons, a new table with essentially the same information was not included.
A conclusive statement related to Ecological function keywords like eutrophication, denitrification, etc. is missing.
Lines 588-604 revised manuscript: The authors have revised the conclusion section to include statements related to ecological function as suggested by the reviewer as follows. ‘…One example would be expanding existing research tools and techniques to assess nutrient dynamics from multiple stacked or integrated CAs to runoff to downstream receiving waterbodies using geographic information system framework with empirically measured denitrification rates across the landscape and into streams, rivers and lakes to map hot spots of nitrogen transport and transformation. Coinciding with this would be assessments of HABs nutrient limitation and nutrient sensitivities to determine how nitrogen affects HABs and possible nitrogen thresholds needed to mitigate HABs. An-other example would be to expand the above efforts to include carbon cycling and aquatic system metabolism using state-of-the-art dissolved oxygen monitoring and models across an empirically measured (e.g., area in ha) gradient of CAs within or among multiple watersheds. A third example would be utilizing and expanding re-search networks across disciplines (engineering, biology, soil science, agronomy, social science) and multiple watersheds at national and international scales. For example, USDA-ARS Long-Term Agroecosystem Research Network (LTAR; https://ltar.ars.usda.gov/) can provide research opportunities to examine the influence of CAs on ecosystem services at large (continental) scales and include scientists from government, academic, and private sectors…’
WATER – 1247233
COMMENTS REVIEWER [3]
Comments and Suggestions for Authors
This study focus on a review analysis of conservation agriculture practices. The study is interesting, however, requires the following revisions.
Abstract: Please give some data related to your research/findings.
Lines 26-41 revised manuscript: The authors note that the journal Water only allows abstract to contain 200 words or less. The abstract was re-written to incorporate the suggested additional data related to the findings of the review paper in the abstract but also not exceed the 200-word total as required for the journal Water as follows.
‘…Conservation agriculture practices (CAs) have been internationally promoted and used for decades to enhance soil health and mitigate soil loss. An additional benefit of CAs has been mitigation of agricultural runoff impacts on aquatic ecosystems. Countries across the globe have agricultural agencies that provide programs for farmers to implement a variety of CAs. Increasingly there is a need to demonstrate that CAs can provide ecological improvements in aquatic ecosystems. Growing global concerns of lost habitat, biodiversity, and ecosystem services, increased eutrophication and associated harmful algal blooms are expected to intensify with increasing global populations and changing climate. We conducted a literature review identifying 88 studies linking CAs to aquatic ecological responses since 2000. Most studies were conducted in North America (78%), primarily the United States (73%), within the framework of the USDA Conservation Effects Assessment Project. Identified studies most frequently documented macroinvertebrate (31%), fish (28%), and algal (20%) responses to riparian (29%), wetland (18%), or combinations (32%) of CAs and/or responses to eutrophication (27%) and pesticide contamination (23%). Notable research gaps include better understanding of biogeochemistry with CAs, quantitative links between varying CAs and ecological responses, and linkages of CAs with aquatic ecosystem structure and function…’
The introduction is short. Improve the content.
The authors have provided more detailed information for the following paragraphs, including added information for some references, to improve the introduction content as follows.
Lines 46-83 revised manuscript: ‘…Healthy aquatic ecosystems are sustainable ecosystems that exhibit resilience in their structure (i.e., biodiversity) and function (i.e., organic matter processing) in response to external stress [1] and subsequently able to provide a variety of ecosystem services including clean water, climate regulation, habitat for plants and animals for wildlife, carbon sequestration, nutrient cycling, and productivity [2-4]. Ecosystem. management approaches that focus on maximizing one ecosystem service result in declines of biodiversity and other ecosystem services [5] Agricultural land use within agricultural watersheds has impacted the ecosystem structure and function of lentic and lotic ecosystems via altered hydrology and increased erosion as a result of land use change and channelization and increased pollution resulting from excess inputs of nutrients, pesticides, and other agricultural contaminants [6]. Subsequently, there is an interest in reducing the impacts of agriculture on aquatic ecosystem structure and function with the use of conservation agriculture practices (CAs) that have been widely implemented in developed regions of the world including the Europe, North America, Asia, and Australia [7-10] through various agricultural agency conservation programs. However, developing countries often have very low CA implementation that greatly limits effective water resource management [11]. Additionally, most agricultural land within the developed regions of the world are privately owned and subsequently the approach towards managing agricultural watersheds differs from that of public wilderness tracts. The management of agricultural watersheds focuses on cropland pro-duction and addressing the subsequent water quality issues via the voluntary adoption by individual landowners of CAs and management of public wilderness tracts focuses on conservation via management by government agencies [3, 12]…’
The authors have provided better description of McPhee et al. [15] by including the following in the Introduction.
Line 90 revised manuscript: ‘…The French Ministry of Agriculture and Food, Territoires d’Innovation projects are agroecosystem living laboratory approaches to enhance innovation in sustainability and resilience to protect soil and biodiversity in agricultural watersheds [15]. Similarly, AAFC also utilizes an agroecosystem living laboratory approach to address agriculturally sourced environmental issues affecting soil and water management and biodiversity with a changing climate. The AAFC living labs projects allow rapid adoption of sustainable practices through close collaboration among researchers, stakeholders and land-use managers (e.g., farmers) [15]…’
The authors have provided better description of Jiao et al. [16] by including the following in the Introduction.
Line 97 revised manuscript: ‘…Within China, Chinese Nationally Important Agricultural Heritage Systems (China – NIAHS) are selected agricultural systems that demonstrate long-term (at least 100 years) sustainable historic agricultural practices providing resilience to extreme conditions (e.g., drought, flooding). These valuable heritage systems provide valuable lessons for biodiversity conservation, soil and water conservation, climate regulation and nutrient cycling while providing food and livelihood security for the rural community [16]…’
The authors have provided better description of Townsend et al. [17] by including the following in the Introduction.
Line 103 revised manuscript: ‘… In Australia, Townsend et al. [17] assessed economic valuation of multiple ecosystem services through tradeoffs between reforestation and agricultural land use as payments for ecosystem services (PES) in a process called ‘bundling’. Ecosystem services via reforestation could include water conservation, carbon sequestration, eco-tourism and conservation of biodiversity. Townsend et al. [17] suggested that the greatest likelihood for success of such a program would need to be through government establishing appropriate mechanisms to subsidize PES payments for water quality improvement.
The authors have provided better description of McDowell et al. [18] by including the following in the Introduction.
Line 110 revised manuscript: ‘…McDowell et al. [18] discussed in detail New Zealand’s water quality policy as outlined in the country’s National Policy Statement on Freshwater Management that requires integrated and sustainable water resource management. New Zealand has a combination of mandatory regulation and voluntary initiatives coinciding with monitoring and evaluation assessment programs to demonstrate successful implementation and regulatory compliance. By comparison, the United Kingdom (UK) has many similar policies and programs as New Zealand. However, unlike New Zealand, the UK utilizes subsidy payments or financial incentives in addition to the previous policies and programs to further encourage stakeholders and land managers to investment in conservation agriculture practices…’
Line 96-99……Revise…….Check “above CA variables”……Use of comma after further…..Check “g) denitrification; and c) denitrification.”
Line 164-167 revised manuscript: The authors have revised the statement from ‘…with CA variables above…’ to ‘…with above CA variables…’ as suggested by the reviewer. The authors have reviewed the syntax and grammar of the sentence and conclude that there is no need to insert a comma after ‘further’. No changes made. The authors agree with the reviewer that denitrification is included twice in the same sentences as both ‘g)’ and ‘c)’ at the end of the sentence. The reference to ‘c) denitrification’ has been deleted and the revised as follows.
From ‘…g) denitrification; and c) denitrification…’ to ‘…and g) denitrification…’
Line 121-122…..The authors can remove full stop in Midwestern U.S. and Southeastern U.S.
Line 191-192 revised manuscript: The authors agree to focus the comparison only with U.S. and other global regions and not focus explicitly on regional locations within the U.S. The sentence was revised as follows. From ‘…within two watersheds in the Midwestern U.S. and one watershed in the Southeastern U.S…’ to ‘…within three watersheds in the U.S…’
Line 149…….Write the scientific names in italics.
Lines 250, 266, 274, 309 revised manuscript: The authors have italicized the scientific name for these organisms.
What is the source of Figure 1?
Lines 131-132 revised manuscript: Figure 1. The authors note that the source of the map is from a U.S. Department of Agriculture, Natural Resources Conservation Service agency website with the following URL:
https://www.nrcs.usda.gov/wps/portal/nrcs/main/national/technical/nra/ceap/ws/
This information has now been included in the Figure 1. legend description below the figure. The map is open access and available to the public and does not require special permission to publish.
The Materials and Methods section may be changed to Methods only.
Line 146 revised manuscript: The authors agree and have changed the section title to ‘Methods’ only as suggested by the reviewer.
Elaborate on the number of studies considered for the study as per the year (at least highest and lowest) in Methods section.
Lines 171-172 revised manuscript: The authors have included additional information on the highest and lowest number of studies considered per year as suggested by the reviewer as follows. ‘…No published studies were observed for the years 2000 and 2018 while the most published studies occurred for the years 2008 and 2012 with 10 and 13 publications, respectively…’
Only two databases were considered??
Lines 151-152 revised manuscript: The authors are aware that there are more than two publication database search engines available. However, some publication database search engines are linked with the two databases used, for example BioOne and the USDA National Agricultural Library (NAL). The authors also used both aforementioned publication database search engines for the current study but did not list these since they both were linked with the two primary publication database search engines used (Google Scholar and SCOPUS). For clarification, the authors have revised the manuscript to include BioOne and NAL as additional publication database search engines used in the Methods section. Other publication database search engines that were available but not used include CORE, that searches only open access research papers and is not broader and more inclusive, and DeepDyve which is a subscription only publication database search engine.
As a result, the authors have revised the ‘Methods’ section detailing the publication database search engines as follows. From ‘…These were obtained through web-based search engines including SCOPUS and Google Scholar…’ to ‘…These were obtained through web-based search engines including SCOPUS, Google Scholar, BioOne, and the U.S. Department of Agriculture National Agricultural Library (NAL)…’
It would be better to also mention the conservation practices taken by the researchers. That would also demarcate the practices that are more suitable in maintaining water quality.
The authors note that this information is clearly reported and discussed in greater detail in the Results and Discussion section of the manuscript along with the reported most frequently assessed conservation practices in the published literature as shown in Figure 4b. For these reasons, the authors have made no additional changes to the manuscript.
The study claims that most relevant studies were conducted in USA. Was there any difference between USA and other countries related to water quality indicators?
The authors note that there were not many differences between the fewer studies in other countries and those conducted within the USA. This indicates that many of the principles of conservation agriculture and biological response paradigms, such as controlling nutrients to control eutrophication, are applicable across broad landscapes. Where the differences do occur are with more local specific economic, climatic, geographical, and biological responses (resulting from genetic diversity). Other differences that might occur have more to do with lack of resources to conduct broader or more in-depth studies in some countries.
Line 311…..Give comma after e.g.
Line 463 revised manuscript: The authors have inserted a comma after ‘e.g.’ as follows ‘e.g.,’ as suggested by the reviewer.
The authors may present their findings in a table on “what CAs are being assessed, and what ecological responses are being measured” for easy understandings.
The authors note that this would seem redundant as this information is provided both in the text (body) of the manuscript as well as in several figures including, but not limited to: Figure 4b (lines 233-236 revised manuscript), new added Figure 5 (lines 272-274 revised manuscript) and revised figures 6a and 6b (lines 271-275 revised manuscript). For these reasons, a new table with essentially the same information was not included.
A conclusive statement related to Ecological function keywords like eutrophication, denitrification, etc. is missing.
Lines 588-604 revised manuscript: The authors have revised the conclusion section to include statements related to ecological function as suggested by the reviewer as follows. ‘…One example would be expanding existing research tools and techniques to assess nutrient dynamics from multiple stacked or integrated CAs to runoff to downstream receiving waterbodies using geographic information system framework with empirically measured denitrification rates across the landscape and into streams, rivers and lakes to map hot spots of nitrogen transport and transformation. Coinciding with this would be assessments of HABs nutrient limitation and nutrient sensitivities to determine how nitrogen affects HABs and possible nitrogen thresholds needed to mitigate HABs. An-other example would be to expand the above efforts to include carbon cycling and aquatic system metabolism using state-of-the-art dissolved oxygen monitoring and models across an empirically measured (e.g., area in ha) gradient of CAs within or among multiple watersheds. A third example would be utilizing and expanding re-search networks across disciplines (engineering, biology, soil science, agronomy, social science) and multiple watersheds at national and international scales. For example, USDA-ARS Long-Term Agroecosystem Research Network (LTAR; https://ltar.ars.usda.gov/) can provide research opportunities to examine the influence of CAs on ecosystem services at large (continental) scales and include scientists from government, academic, and private sectors…’

Reviewer 4 Report
General comment:
The paper seems well written with various concepts. But I am adding a moderate revision as there is much more ample scope to include. The authors have discussed the synthesis of the literature from 2000-2020 delineated that research on aquatic ecological response, but I am not able to see any good discussion. Please try to address the following issues:
(i) how Conservation Agriculture Practices e enable to mitigate climate change?;
(ii) how feedback work between relationships between eutrophication, nutrient source improved understanding of how are stream and lake nutrient spiralling, cycling, and internal loading (retention) are affected by CAs?;
(iii) a better understanding of the effect of CAs on physical habitat quality as well as water quality,
(iv)) clarifying how and to what extent stream and lake trophic states and interactions are altered with changes in water quality?
Specific comments:
In lines “study on mammals, specifically beaver (Castor fiber) “ use italics for the species name.
I propose to correct the following publications:
Please describe these works better and more precisely:
McPhee, C.; Bancerz, M.; Mambrini-Doudet, M.; Chrétien, F.; Huyghe, C.; Gracia-Garza, J. The defining characteristics of agroecosystem living Labs. Sustainability 2021, 13, 1718. https://doi.org/10.3390/su13041718 503 16. Jiao, W.; Min, Q. Reviewing the progress in the identification, conservation and management of China-Nationally important agricultural heritage systems (China-NIAHS). Sustainability 2017, 9, 1698. doi.org/10.3390/su9101698 505 17. Townsend, P.V.; Harper, R.J.; Brennan, P.D.; Dean, C.; Wu, S.; Smettem, K.R.J.; Cook, S.E. Multiple environmental services as an opportunity for watershed restoration. Forest Policy Econ 2012, 17, 45–58. doi.org/10.1016/j.forpol.2011.06.008 507 18. McDowell, R.W.; Dils, R.M.; Collins, A.L.; Flahive, K.A.; Sharpley, A.N.; Quinn, J. A review of the policies and implementation of practices to decrease water quality impairment by phosphorus in New Zealand, the UK, and the US. Nutr Cycl Agroecosyst 509 2016, 104, 289–305. doi.org/10.1007/s10705-015-9727-0
Please add more information to the text. Try your best:
Christensen, V.G.; Lee, K.E.; McLees, J.M.; Niemela, S.L. Relations between retired agricultural land, water quality, and aquatic community health, Minnesota River Basin. J Environ Qual 2012, 41, 1459-1472. doi.org/10.2134/jeq2011.0468 546 32. Chu, C.; Minns, C.K.; Lester, N.P.; Mandrak, N.E. An updated assessment of human activities, the environment, and freshwater fish biodiversity in Canada. Can J Fish Aquat Sci 2015, 72, 135–148. dx.doi.org/10.1139/cjfas-2013-0609 548 33. Cullum, R.F.; Knight, S.S.; Cooper, C.M.; Smith, S. 2006. Combined effects of best management practices on water quality in oxbow lakes from agricultural watersheds. Soil Till Res 2006, 90, 212-221. doi.org/10.1016/j.still.2005.09.004
Get more data from these publications:
Goeller, B.C.; Febria, C.M.; McKergow, L.A.; Harding, J.S. Combining tools from edge-of-field to in-stream to attenuate reactive nitrogen along small agricultural waterways. Water 2020, 12, 383. doi.org/10.3390/w12020383
Broadmeadow, S.; Nisbet, T.R. The effects of riparian forest management on the freshwater environment: a literature review of best management practice. Hydrol Earth Sys Sci 2004, 8, 286-305.
You worded that:
“These results from Ullah et al. [93] indicated that differences in land use leads to differences in the watershed’s capacity for denitrification and is a direct result of different physicochemical properties of the two soil types where wetter soils, greater soil porosity, and greater soil organic carbon loads found in forested riparian soils significantly increase denitrification compared with cultivated soils. In addition, many of these same soil properties also allow greater N mineralization and lower levels of N2O emission in forested riparian soils further highlighting the value of using forested riparian buffers as a CA practice”. But, this supported only by one publication. Try to add more related papers.
Detailed remarks:
-
Channelized agricultural headwater streams (i.e., agricultural drainage ditches) are common in agricultural watersheds in the Midwestern United States, Canada, and Europe. Please try to add more than two sentences.
-
Implemented CA practices include conservation tillage practices, specialized drainage culverts to reduce runoff flow rates is important. Planting of temporally stable vegetated buffers and grassed filters, conversion of highly erodible cropland to vegetative cover now is a high priority. Discus more the influences of changing nutrient-pesticide mixtures on the biota as a result of water quality improvements due to implementation of CAs. I suggest reviewing more article on different model outputs how the hydrology-green cover-climate change loop works. How green cover and the blue cover is contributing to climate change? Add more new publishers who are currently discussing quantifying agricultural watersheds that need to receive an implementation of documentation of which combinations of CAs provide the greatest ecological benefits.
-
Can you add a graph, picture or flow chart for constructed wetland, wildlife habitat vegetative buffers, and a two-celled sediment retention pond?
-
This text is not supported by any publication“Ecological functional responses include physical processes of hydrology, morphology, and physical and chemical components that affect ecosystem responses. Aspects of biogeochemistry including uptake, transport, and storage of nutrients are also important elements of lotic and lentic aquatic ecosystems that can assist with understanding ecosystem responses to agricultural conservation practices. To date, ecological research in agricultural watersheds has encompassed a broad suite of ecological responses scaling along the hierarchy of biological organization including nitrogen biogeochemistry, microbial ecology of phytoplankton and heterotrophic bacteria, a multitude of ecotoxicological assessments ranging from standard toxicity bioassays to biomarker assessments, and fish ecology assessments involving population and community ecology. However, most studies are conducted using comparisons of systems across a gradient of CA and/or stressor conditions such as reference watershed versus agriculturally impacted watershed ecological responses” or am I wrong?
-
In the Conclusion section you commented:
“Particularly, new research efforts involving the individual (e.g., genetic, biochemical, physiological) and population level (e.g., reproductive output) impacts of agricultural contaminants would assist with understanding the sublethal impacts of agricultural contaminants and the potential benefits of water quality improvements beyond the current regulatory standards”. However, you have provided too little data on this issue.
-
Novel research that documents the influence of CAs on ecosystem structure and function within and among watersheds would be useful to further our understanding of whether conservation practices can restore aquatic ecosystems. This is an interesting approach. Please add more in the Discussion section.
-
Closing these information gaps will provide landowners and stakeholders with better management plans and tools to more efficiently maximize the benefits and ecosystem services of implementing CAs within agricultural watersheds. Ecosystem services are a very broad topic, and not were found in the text. Why you have applied this issue in the Conclusion section? Please delete or add more to the review paper.
Author Response
WATER – 1247233
COMMENTS REVIEWER [4]
Comments and Suggestions for Authors
General comment:
The paper seems well written with various concepts. But I am adding a moderate revision as there is much more ample scope to include. The authors have discussed the synthesis of the literature from 2000-2020 delineated that research on aquatic ecological response, but I am not able to see any good discussion. Please try to address the following issues:
(i) how Conservation Agriculture Practices e enable to mitigate climate change?;
The authors note that CAs are likely to have some ameliorating effects on climate change, however the extent to which such practices will do this is unknown in the literature and beyond the scope of the current literature review. Such research is the focus of some current studies by researchers within the USDA-ARS, specifically the Long-Term Agroecosystem Research Network (LTAR) as provided in the following link: https://ltar.ars.usda.gov/. Measurements of carbon sequestration by riparian buffers and/or cover crops and reductions in greenhouses gasses resulting from CAs at the watershed scale is beginning to accelerate through the use of equipment such as networking of eddy covariance gas flux towers but the research is still in its infancy. However, several studies in the last two decades have attempted to ascertain how climate change may impact the effectiveness of CAs, themselves, and how such influences will coincidentally affect ecological responses. The authors agree that climate change needs to be better addressed within the Results and Discussion section of the manuscript and several studies included in the review do address global climate change as an issue affecting biodiversity and how CAs may or may not be effective at mitigating agricultural impacts within a changing climate. For these reasons, the authors have included a paragraph addressing climate change relative to CAs and biodiversity in Section 3.3 Structural Ecological Responses as follows.
Lines 391-408 revised manuscript: added ‘…Global climate change can have long-term and watershed-scale influences on CA effectiveness in mitigating transport of agricultural pollutants to streams and lakes. Collectively, these multiple stressors can alter the health and diversity of aquatic ecosystems and fish communities [32,39,45,76,77]. Recent research has attempted to address how expected global climate change might affect long-term effectiveness of present-day CAs to try and reduce uncertainty and protect water quality and fish biodiversity with conservation planning [39]. Recent research by Fraker et al. [39] and Hall et al. [45] in the U.S. and Canada integrated the use of several models to attempt to forecast global climate change scenarios to estimate CA effectiveness and resiliency under altered precipitation patterns relative to protecting fish community diversity. The studies indicated that changes in climate are likely to increase the need for CAs to protect diversity [45], but that efforts to improve water quality through CAs may could come with costs to other ecosystem services such as unintended shifts fish communities [39]. Studies by Sarkar et al. [76,77] in India indicated that fish in the region were maturing earlier, due in part, to warming temperatures from climate change and alterations in hydrology from human activity such as irrigation for agriculture. As a result, Sarkar et al. [77] emphasized the need to maintain and enhance riparian floodplain wetland conservation as a means to ameliorate the impacts of agriculture and climate change to protect fish diversity in watersheds of India…’
- Chu, C.; Minns, C.K.; Lester, N.P.; Mandrak, N.E. An updated assessment of human activities, the environment, and freshwater fish biodiversity in Canada. Can J Fish Aquat Sci 2015, 72, 135–148. dx.doi.org/10.1139/cjfas-2013-0609
- Fraker, M.E.; Keitzer, S.C.; Sinclair, J.S.; Aloysius, N.R.; Dippold, D.A.; Yen, H. Projecting the effects of agricultural conser-vation practices on stream fish communities in a changing climate. Sci Tot Environ 2020, 747, 141112. https://doi.org/10.1016/j.scitotenv.2020.141112
- Hall, K.R.; Herbert, M.E.; Sowa, S.P.; Mysorekar, S.; Woznicki, S.A.; Nejadhashemi, P.A.; Wang, L. Reducing current and future risks: Using climate change scenarios to test an agricultural conservation framework. J Great Lakes Res 2017, 43, 59-68. http://dx.doi.org/10.1016/j.jglr.2016.11.005
- Sarkar, U.K.; Pathak, A.K.; Sinha, R.K.; Sivakumar, K.; Pandian, A.K.; Pandey, A.; Dubey, V.K.; Lakra, W.S. Freshwater fish biodiversity in the River Ganga (India): changing pattern, threats and conservation perspectives. Rev Fish Biol Fisheries 2012, 22, 251–272. doi.org/10.1007/s11160-011-9218-6
- Sarkar, U.K.; Bakshi, S.; Lianthuamluaia, L.; Mishal, P.; Das Ghosh, B.; Saha, S.; Karnatak, G. Understanding enviro‑climatological impact on fish biodiversity of the tropical floodplain wetlands for their sustainable management. Sustain Wat Resource Manage 2020, 6, 96. https://doi.org/10.1007/s40899-020-00445-0
(ii) how feedback work between relationships between eutrophication, nutrient source improved understanding of how are stream and lake nutrient spiralling, cycling, and internal loading (retention) are affected by CAs?;
The authors agree that understanding the relationships and interactions between CAs and eutrophication, external and internal nutrient loadings, and nutrient cycling and spiraling in lakes and streams is very valuable in determining the effectiveness of CAs at controlling eutrophication in aquatic ecosystems. This has been one of the goals for researchers attempting to demonstrate that CAs are capable of long-term improvement of water quality improvement aquatic ecosystem health at the watershed scale. This outcome has also been one of the most challenging to measure and one of the primary reasons for the authors for constructing this review was an attempt to ascertain our current understanding of how CAs can or cannot effectively protect water quality and aquatic ecosystem health long-term. This also includes how CAs affect ecological functions such as nutrient cycling, spiraling, internal loading, nutrient availability, nutrient limitation, and eutrophication eliciting harmful algal blooms in streams, rivers and lakes. That the available empirical literature is limited is indicative of how much is still unknown and in need of a better understanding. New empirical research addressing many of these questions and issues is currently underway and is only beginning to be published, but many recent research papers focus on smaller scale CA measures (e.g., experimental wetlands, farm fields, drainage ditches, etc.) and are not yet being conducted at a watershed scale. The authors agree that eutrophication and nutrient dynamics needs to be more thoroughly discussed in the current review paper, specifically within the Results and Discussion section, sub-section 3.4 Functional Ecological Responses and expand on the current paragraph discussing eutrophication. For these reasons, the authors have expanded the paragraph addressing eutrophication relative to CAs to include a discussion of nutrient dynamics as follows.
Lines 438-453 revised manuscript: added ‘…Nutrient dynamics in streams, rivers, and lakes of agricultural watersheds are often linked to upland terrestrial legacy nutrients such as phosphorus [6,99] and/or subsurface transport of nitrogen, often as nitrate [14,43,44]. This increase in nutrients can then alter transport and fate, increase productivity, and potentially elicit HABs [44,48]. While alteration of nutrient dynamics can occur within the lotic or lentic system [14,48,65], CAs can be utilized to harness some watershed ecosystem functions [6,14,43,99]. Denitrification and phosphorus sequestration (e.g., plant uptake) that mitigate alteration of nutrient dynamics. The framework proposed by Goeller et al. [43] using multiple stacked CAs combined with in-stream management to enhance ecological processes such as denitrification could be used to control nutrients such as nitrogen. Also, managing channelized agricultural headwater streams for both drainage (i.e., drainage ditches) and enhanced ecological processes can be enlisted to further control excess nutrients that alter nutrient dynamics in streams, rivers and lakes [14]. Currently, a research gap remains between measured reductions in nutrients with CAs and how these reductions shift or alter nutrient dynamics in impacted agricultural watershed water bodies and future research is needed…’
- Withers, P.J.A.; Vadas, P.A.; Uusitalo, R.; Forber, K.J.; Hart, M.; Foy, R.H.; Delgado, A.; Dougherty, W.; Lilja, H.; Burkitt, L.L.; Rubæk, G.H.; Pote, D.; Barlow, K.; Rothwell, S.; Owens, P.R. A global perspective on integrated strategies to manage soil phosphorus status for eutrophication control without limiting land productivity. J Environ Qual 2019, 48, 1234–1246. doi.org/10.2134/jeq2019.03.0131
- Pierce, S.C.; Kröger, R.; Pezeshki, R. Managing artificially drained low-gradient agricultural headwaters for enhanced eco-system functions. Biology 2012, 1, 794-856. doi.org/10.3390/biology1030794
- Goeller, B.C.; Febria, C.M.; McKergow, L.A.; Harding, J.S. Combining tools from edge-of-field to in-stream to attenuate reactive nitrogen along small agricultural waterways. Water 2020, 12, 383. doi.org/10.3390/w12020383
- Gilbert, P.M. From hogs to HABs: impacts of industrial farming in the US on nitrogen and phosphorus and greenhouse gas pollution. Biogeochemistry 2020, 150, 139–180. https://doi.org/10.1007/s10533-020-00691
- James, R.T.; Havens, K.E.; McOrmick, P.; Jones, B.; Ford. C. Water quality trends in shallow South Florida lakes and assessment of regional versus local forcing functions. Crit Rev Environ Sci Technol 2011, 41:S1, 576-607. doi.org/10.1080/10643389.2010.530581
- Lürling, M.; Mucci, M. Mitigating eutrophication nuisance: in-lake measures are becoming inevitable in eutrophic waters in the Netherlands. Hydrobiologia 2020, 847, 4447–4467. https://doi.org/10.1007/s10750-020-04297-9
- Withers, P.J.A.; Neal, C.; Jarvie, H.P.; Doody, D.G. Agriculture and eutrophication: Where do we go from here? Sustainability 2014, 6, 5853-5875. doi.org/10.3390/su6095853
(iii) a better understanding of the effect of CAs on physical habitat quality as well as water quality,
As suggested by the reviewer, the authors have revised section 3.1 (lines 180-193 revised manuscript) and added a section discussing the impacts of CRP and CREP riparian buffers on physical habitat quality and indicated a need for more research on this topic. Additionally, in section 3.3 (line 323) the authors confirmed that CAs of grass riparian buffers and atrazine reduction practices did not improve physical habitat quality or water quality. Revised as follows.
Lines 219-232 revised manuscript: ‘…Christensen et al. [31] documented that percentage agricultural land retirement through Conservation Reserve Program was positively correlated with fish community structure at smaller reach scales, but not at the watershed scale in the Minnesota River basin in Minnesota. Additionally, Christensen et al. [34] suggested their fish-habitat relationships suggested that land retirement adjacent to streams may result in improved physical habitat qual-ity, but they did not directly assess the relationships between percentage of land retirement and physical habitat quality. The influence of CAs on physical habitat quality has not been widely assessed. Smiley et al. [82, 84] found that herbaceous riparian buffers planted as part of the Conservation Reserve Enhancement Program adjacent to channelized agricultural headwater streams in Ohio only widen the riparian habitats and did not alter vegetative type, vegetative structure, or instream habitat variables. The linkages between CAs and physical habitat quality have not been widely explored because most CAs are designed to improve water quality, not physical habitat quality and as such this is an area that future research needs to explore in more detail…’
Line 365-366 revised manuscript: ‘…improved water quality conditions or physical habitat quality within these streams…’
(iv)) clarifying how and to what extent stream and lake trophic states and interactions are altered with changes in water quality?
There is not much literature on this. Most work with attempts to alter trophic states through management (within river/lake system or externally) has been done in Europe as descried in detail by Jeppesen et al. (2207a; 2007b and Scheffer (2001; 2004). These European studies did not specifically address the ability of CAs to shift or alter trophic status, but rather to address broader turbidity issues arising from poor water quality and increased eutrophication with attempts to control eutrophication and restore/rehabilitate the system to a less turbid ‘clear water’ trophic state. Currently, the few studies that have attempted to link CAs or land-use changes in agricultural watersheds to trophic state include a published review by Cook (2007) for North American Lakes, work by Poor (2010) in Florida, work by Thomatou et al. (2013) in Greece, and a recent publication by Lizotte et al. 2021 in a special issue for the journal Water assessing responses of trophic states in a lake with multiple CAs integrated over time (Beasley Lake). However, the research by Poor (2010) only qualitatively assesses CAs by examining lake trophic state after establishment of a constructed wetland. Thomatou et al. (2013) only assessed trophic state relative to broader watershed land-use changes and not any specific CAs. Research reported by Lizotte et al. (2021) is outside the scope of the current paper as the Methods listed research through 2020 and this is a more recent publication. As demonstrated above, there is a paucity of literature available prior to 2021 that clearly and empirically links CAs with interactions and alterations in trophic states and the authors emphasized this, appropriately, in the Conclusion section where notable information gaps exist. For the reasons outlined above, the authors have made no changes to the manuscript.
Cooke, G.D. History of eutrophic lake rehabilitation in North America with arguments for including social sciences in the paradigm. Lake Reserv. Manag. 2007, 23, 323–329.
Jeppesen, E., M. Søndergaard, T. L. Lauridsen, B. Kronvang, M. Beklioglu, E. Lammens, H. S. Jensen, J. Köhler, A-M. Ventelä, M. Tarvainen and I. Tátrai (2007a): Danish and other European experiences in managing shallow lakes, Lake and Reservoir Management, 23:4, 439-451. http://dx.doi.org/10.1080/07438140709354029
Jeppesen, E., Søndergaard, M., Meerhoff, M., Lauridsen, T.L., Jensen, J.P. (2007b) Shallow lake restoration—Some recent findings and challenges ahead. Hydrobiologia, 584, 239–252.
Lizotte, R.E., Jr.; Yasarer, L.M.W.; Bingner, R.L.; Locke, M.A.; Knight, S.S. Long-Term Oxbow Lake Trophic State under Agricultural Best Management Practices. Water 2021, 13, 1123. https://doi.org/10.3390/
w13081123
Poor, N.D. Effect of lake management efforts on the trophic state of a subtropical shallow lake in Lakeland, Florida, USA. Water Air Soil Pollut. 2010, 207, 333–347.
Scheffer, M. (2001) Alternative attractors of shallow lakes. TheScientificWorld (2001) 1, 254–263
Scheffer, M. (2004) Ecology of Shallow Lakes. Springer
Thomatou, A.A.; Triantafyllidou, M.; Chalkia, E.; Kehayias, G.; Konstantinou, I.; Zacharias, I. Land use changes do not rapidly change the trophic state of a deep lake. Amvrakia Lake, Greece. J. Environ. Prot. 2013, 4, 426–434.
Specific comments:
In lines “study on mammals, specifically beaver (Castor fiber) “ use italics for the species name.
Line 309 revised manuscript: The authors have italicized the scientific name for beaver (Castor fiber).
I propose to correct the following publications:
Please describe these works better and more precisely:
McPhee, C.; Bancerz, M.; Mambrini-Doudet, M.; Chrétien, F.; Huyghe, C.; Gracia-Garza, J. The defining characteristics of agroecosystem living Labs. Sustainability 2021, 13, 1718. https://doi.org/10.3390/su13041718 503 16.
The authors have provided better description of McPhee et al. [15] by including the following in the Introduction.
Line 90 revised manuscript: ‘…The French Ministry of Agriculture and Food, Territoires d’Innovation projects are agroecosystem living laboratory approaches to enhance innovation in sustainability and resilience to protect soil and biodiversity in agricultural watersheds [15]. Similarly, AAFC also utilizes an agroecosystem living laboratory approach to address agriculturally sourced environmental issues affecting soil and water management and biodiversity with a changing climate. The AAFC living labs projects allow rapid adoption of sustainable practices through close collaboration among researchers, stakeholders and land-use managers (e.g., farmers) [15]…’
Jiao, W.; Min, Q. Reviewing the progress in the identification, conservation and management of China-Nationally important agricultural heritage systems (China-NIAHS). Sustainability 2017, 9, 1698. doi.org/10.3390/su9101698 505 17.
The authors have provided better description of Jiao et al. [16] by including the following in the Introduction.
Line 97 revised manuscript: ‘…Within China, Chinese Nationally Important Agricultural Heritage Systems (China – NIAHS) are selected agricultural systems that demonstrate long-term (at least 100 years) sustainable historic agricultural practices providing resilience to extreme conditions (e.g., drought, flooding). These valuable heritage systems provide valuable lessons for biodiversity conservation, soil and water conservation, climate regulation and nutrient cycling while providing food and livelihood security for the rural community [16]…’
Townsend, P.V.; Harper, R.J.; Brennan, P.D.; Dean, C.; Wu, S.; Smettem, K.R.J.; Cook, S.E. Multiple environmental services as an opportunity for watershed restoration. Forest Policy Econ 2012, 17, 45–58. doi.org/10.1016/j.forpol.2011.06.008 507 18.
The authors have provided better description of Townsend et al. [17] by including the following in the Introduction.
Line 103 revised manuscript: ‘… In Australia, Townsend et al. [17] assessed economic valuation of multiple ecosystem services through tradeoffs between reforestation and agricultural land use as payments for ecosystem services (PES) in a process called ‘bundling’. Ecosystem services via reforestation could include water conservation, carbon sequestration, eco-tourism and conservation of biodiversity. Townsend et al. [17] suggested that the greatest likelihood for success of such a program would need to be through government establishing appropriate mechanisms to subsidize PES payments for water quality improvement.
McDowell, R.W.; Dils, R.M.; Collins, A.L.; Flahive, K.A.; Sharpley, A.N.; Quinn, J. A review of the policies and implementation of practices to decrease water quality impairment by phosphorus in New Zealand, the UK, and the US. Nutr Cycl Agroecosyst 509 2016, 104, 289–305. doi.org/10.1007/s10705-015-9727-0
The authors have provided better description of McDowell et al. [18] by including the following in the Introduction.
Line 110 revised manuscript: ‘…McDowell et al. [18] discussed in detail New Zealand’s water quality policy as outlined in the country’s National Policy Statement on Freshwater Management that requires integrated and sustainable water resource management. New Zealand has a combination of mandatory regulation and voluntary initiatives coinciding with monitoring and evaluation assessment programs to demonstrate successful implementation and regulatory compliance. By comparison, the United Kingdom (UK) has many similar policies and programs as New Zealand. However, unlike New Zealand, the UK utilizes subsidy payments or financial incentives in addition to the previous policies and programs to further encourage stakeholders and land managers to investment in conservation agriculture practices…’
Please add more information to the text. Try your best:
Christensen, V.G.; Lee, K.E.; McLees, J.M.; Niemela, S.L. Relations between retired agricultural land, water quality, and aquatic community health, Minnesota River Basin. J Environ Qual 2012, 41, 1459-1472. doi.org/10.2134/jeq2011.0468 546 32.
The authors have provided more information for Christensen et al. [31] in the manuscript by including the following in the Results and Discussion section.
Lines 219-232 revised manuscript: added ‘…Christensen et al. [31] documented that percentage agricultural land retirement through Conservation Reserve Program was positively correlated with fish community structure at smaller reach scales, but not at the watershed scale in the Minnesota River basin in Minnesota. Additionally, Christensen et al. [34] suggested their fish-habitat relationships suggested that land retirement adjacent to streams may result in improved physical habitat qual-ity, but they did not directly assess the relationships between percentage of land retirement and physical habitat quality. The influence of CAs on physical habitat quality has not been widely assessed. Smiley et al. [82, 84] found that herbaceous riparian buffers planted as part of the Conservation Reserve Enhancement Program adjacent to channelized agricultural headwater streams in Ohio only widen the riparian habitats and did not alter vegetative type, vegetative structure, or instream habitat variables. The linkages between CAs and physical habitat quality have not been widely explored because most CAs are designed to improve water quality, not physical habitat quality and as such this is an area that future research needs to explore in more detail…’
Chu, C.; Minns, C.K.; Lester, N.P.; Mandrak, N.E. An updated assessment of human activities, the environment, and freshwater fish biodiversity in Canada. Can J Fish Aquat Sci 2015, 72, 135–148. dx.doi.org/10.1139/cjfas-2013-0609 548 33.
The authors have provided more information for Chu et al. [32] in the manuscript by including the following in the Results and Discussion section.
Lines 194-202 revised manuscript: added ‘…Chu et al. [32] in Canada used modeling scenarios to assess potential fish diversity responses to changes in climate and land management resulting from climate change and associated land-use changes. The study provided modeled scenarios indicating conservation management priorities, including use of CAs, would most likely increase for watersheds along the eastern and western maritime coastlines to maintain fish diversity in these regions. Chu et al. [32] concluded that, ‘To be effective, conservation and management of aquatic habitats and resources should aim to keep pace with changes in the types and concentration of human activities and environmental change across the landscape.’…’
Cullum, R.F.; Knight, S.S.; Cooper, C.M.; Smith, S. 2006. Combined effects of best management practices on water quality in oxbow lakes from agricultural watersheds. Soil Till Res 2006, 90, 212-221. doi.org/10.1016/j.still.2005.09.004
The authors have provided more information for Cullum et al. [33] in the manuscript by including the following in the Results and Discussion section.
Lines 515-519 revised manuscript: added ‘…Cullum et al. [33] did not observe any effects of CAs on either coliform or enterococci bacteria counts. However, algal biomass, as chlorophyll concentration, clearly increased four years after implementation of CAs coinciding with decreases in suspended sediments indicating algal populations were primarily light limited but not nutrient limited [33]…’
Get more data from these publications:
Goeller, B.C.; Febria, C.M.; McKergow, L.A.; Harding, J.S. Combining tools from edge-of-field to in-stream to attenuate reactive nitrogen along small agricultural waterways. Water 2020, 12, 383. doi.org/10.3390/w12020383
The authors have provided more information for Goeller et al. [43] in the manuscript by including the following in the Results and Discussion section.
Lines 454-463 revised manuscript: added ‘…Biogeochemical processes can be harnessed within a framework of integrated or stacked CAs and stream habitat restoration to improve nutrient cycling and increase nutrient attenuation in agricultural watersheds. Goeller et al. [43] proposed a framework of stacked CAs including edge-of-field practices (e.g., grassed filter strips, retention ponds), riparian buffers (e.g., constructed wetlands) combined with in-stream habitat structures (e.g., woody debris, low-grade weirs) to enhance biogeochemical attenuation of nitrogen in New Zealand agricultural watersheds. Goeller et al.’s [43] framework is flexible to allow researchers, stakeholders, land-owners and farmers a variety of options that are field to farm-specific and increase the likelihood for success in assessing the most effective measures for mitigating farm nutrient runoff…’
Broadmeadow, S.; Nisbet, T.R. The effects of riparian forest management on the freshwater environment: a literature review of best management practice. Hydrol Earth Sys Sci 2004, 8, 286-305.
The authors have provided more information for Broadmeadow et al. [10] in the manuscript by including the following in the Introduction section.
Lines 207-215 revised manuscript: following the text ‘…Globally, riparian (29.5%) and wetland (18.2%) CAs were the most frequently assessed individually for aquatic ecological responses…’ more information is provided as ‘…A literature review of riparian forest CAs by Broadmeadow and Nisbet [10] assessed how riparian functions are affected by design and management of the CAs. Specifically, Broadmeadow and Nisbet [10] reviewed how riparian width and vegetative structure and species could protect aquatic ecosystems within silviculture watersheds. In general riparian widths near 30 m provided greater protection for water quality and greater ecosystem services such as denitrification, habitat, native animal and plant diversity, temperature moderation, and sediment removal…’
You worded that:
“These results from Ullah et al. [93] indicated that differences in land use leads to differences in the watershed’s capacity for denitrification and is a direct result of different physicochemical properties of the two soil types where wetter soils, greater soil porosity, and greater soil organic carbon loads found in forested riparian soils significantly increase denitrification compared with cultivated soils. In addition, many of these same soil properties also allow greater N mineralization and lower levels of N2O emission in forested riparian soils further highlighting the value of using forested riparian buffers as a CA practice”. But, this supported only by one publication. Try to add more related papers.
The authors have included another related paper by Keating et al. (2016) that examined Lines 320-324
Lines 477-485 revised manuscript: added ‘…Similarly, Keating et al. [94] measured soil denitrification gene (nosZ) abundances and N2O emissions in CAs and cropland within Beasley Lake Watershed in Mississippi during summer 2013. Keating et al. [94] observed highest N2O in cropland and riparia, although measured soil genetic and chemical data suggested a difference in N2O sources between these two sites. Denitrification gene (nosZ) abundances in soils indicated denitrification-derived N2O products in intermittently flooded wetland and riparia sites, while soil inorganic nitrogen concentrations indicated greater nitrification-derived N2O products in dry cropland sites in the watershed…’
New Reference:
[94] Keating, M.P.; Ochs, C.A.; Balachandran, D.; Holland, M.M.; Lizotte, R.E.; Yu. K. Spatial variation related to hydrologic patterns and vegetation in greenhouse gas fluxes from the Mississippi Delta agricultural region. Appl Soil Ecol 2016, 98, 278-281. http://dx.doi.org/10.1016/j.apsoil.2015.09.012
Channelized agricultural headwater streams (i.e., agricultural drainage ditches) are common in agricultural watersheds in the Midwestern United States, Canada, and Europe. Please try to add more than two sentences.
As suggested by the reviewer, the authors added additional description of the characteristic geomorphology and riparian habitat characteristics of channelized agricultural headwater streams. The manuscript is revised as follows.
Lines 246-249 revised manuscript: added ‘…These are first to third order streams that have been modified or created for agricultural drainage [13]. Channelized agricultural headwater streams are characterized by trapezoidal, straightened, widened over enlarged channels dominated by herbaceous vegetation [13, 82]…’
Implemented CA practices include conservation tillage practices, specialized drainage culverts to reduce runoff flow rates is important. Planting of temporally stable vegetated buffers and grassed filters, conversion of highly erodible cropland to vegetative cover now is a high priority. Discus more the influences of changing nutrient-pesticide mixtures on the biota as a result of water quality improvements due to implementation of CAs. I suggest reviewing more article on different model outputs how the hydrology-green cover-climate change loop works. How green cover and the blue cover is contributing to climate change? Add more new publishers who are currently discussing quantifying agricultural watersheds that need to receive an implementation of documentation of which combinations of CAs provide the greatest ecological benefits.
The authors note that the focus of the review paper is not to discuss the numerous modeled outputs of CAs on improving water quality under changing climate conditions as there are already several available that already do this. For example, recent papers by Yasarer et al. (2017) and Osmond (2019) address some of these climate change-water quality-CAs interactions. However, these do not focus on ecological endpoints, only implying if agricultural contaminants in runoff continue to be effectively controlled under current CAs within a changing climate scenario, then there would be concomitant (but unmeasured) ecological benefits. This paper primarily examines the literature on measured ecological benefits of CAs in aquatic ecosystems. While there are some papers that are highlighted in this review where models were used to attempt to ascertain previously measured ecological benefits in aquatic systems within agricultural watersheds (see Chu et al. 2015; Fraker et al. 2020), adding additional literature that is primarily examining water quality (nutrients, pesticides, suspended sediments, etc.) with no clear measured ecological endpoints is beyond the scope of the current review. There are only a few papers that have assessed ecological responses to changes in more complex nutrient-pesticide mixtures (see Lizotte et al. 2012; Lizotte et al. 2014; Lizotte and Moore 2017). Although Lizotte et al. 2012 studied these effects within an agricultural watershed context, Lizotte et al. 2014 and Lizotte and Moore 2017 studied these effects exclusively in various scaled experimental constructed wetlands in a minimally impacted forested watershed. For this reason, the Lizotte et al. 2014 and Lizotte and Moore 2017 studies was not included in this review.
Chu, C.; Minns, C.K.; Lester, N.P.; Mandrak, N.E. An updated assessment of human activities, the environment, and freshwater fish biodiversity in Canada. Can J Fish Aquat Sci 2015, 72, 135–148. dx.doi.org/10.1139/cjfas-2013-0609
Fraker, M.E.; Keitzer, S.C.; Sinclair, J.S.; Aloysius, N.R.; Dippold, D.A.; Yen, H. Projecting the effects of agricultural conser-vation practices on stream fish communities in a changing climate. Sci Tot Environ 2020, 747, 141112. https://doi.org/10.1016/j.scitotenv.2020.141112
Lizotte, R.E.; Shields, F.D.; Murdock, J.N.; Knight, S.S. Responses of Hyalella azteca and phytoplankton to a simulated agricultural runoff event in a managed backwater wetland. Chemosphere 2012, 87, 684–691. doi.org/10.1016/j.chemosphere.2011.12.058
Lizotte Jr, R.E., M. A. Locke & S. Testa III (2014) Influence of varying nutrient and pesticide mixtures on abatement efficiency using a vegetated free water surface constructed wetland mesocosm, Chemistry and Ecology, 30:3, 280-294, DOI:10.1080/02757540.2013.861823
Lizotte Jr, R.E. & M. T. Moore (2017) Effectiveness of emergent and submergent aquatic plants in mitigating a nitrogen–permethrin mixture, Chemistry and Ecology, 33:5, 420-433, DOI: 10.1080/02757540.2017.1310849
Osmond, D. L., A. L. Shober, A. N. Sharpley, E. W. Duncan, and D. L. K. Hoag (2019) Increasing the effectiveness and adoption of agricultural phosphorus management strategies to minimize water quality impairment. J. Environ. Qual. 48:1204–1217. doi:10.2134/jeq2019.03.0114
Yasarer, L. M. W., R. L. Bingner, J. D. Garbrecht, M. A. Locke, R. E. Lizotte, Jr., H. G. Momm, P. R. Busteed (2017) Climate change impacts on runoff, sediment, and nutrient loads in an agricultural watershed in the Lower Mississippi River Basin. Applied Engineering in Agriculture 33(3): 379-392, https://doi.org/10.13031/aea.12047
Can you add a graph, picture or flow chart for constructed wetland, wildlife habitat vegetative buffers, and a two-celled sediment retention pond?
Line 271 revised manuscript: Following the brief description of Beasley Lake Watershed in western Mississippi, USA at line 271 in the revised manuscript, the authors have included a new Figure 5 and legend as suggested by the reviewer. ‘…Figure 5. Beasley Lake Watershed in western Mississippi, USA with implemented Conservation Agriculture practices: veg-etated buffers and grassed filters (VBS); conservation reserve program conversions (CRP); constructed wetland (CW); wild-life habitat vegetated buffers to attract bobwhite quail (Colinus virginianus) (QB); and a two-celled sediment retention pond (SP)…’
Figures following the new Figure 5 have now been renumbered as Figure 6 a-b and Figure 7 and revised throughout the manuscript when referenced.
This text is not supported by any publication “Ecological functional responses include physical processes of hydrology, morphology, and physical and chemical components that affect ecosystem responses. Aspects of biogeochemistry including uptake, transport, and storage of nutrients are also important elements of lotic and lentic aquatic ecosystems that can assist with understanding ecosystem responses to agricultural conservation practices. To date, ecological research in agricultural watersheds has encompassed a broad suite of ecological responses scaling along the hierarchy of biological organization including nitrogen biogeochemistry, microbial ecology of phytoplankton and heterotrophic bacteria, a multitude of ecotoxicological assessments ranging from standard toxicity bioassays to biomarker assessments, and fish ecology assessments involving population and community ecology. However, most studies are conducted using comparisons of systems across a gradient of CA and/or stressor conditions such as reference watershed versus agriculturally impacted watershed ecological responses” or am I wrong?
The authors thank the reviewer for bringing this oversight to the authors attention. The statements in the text should have been supported by several publications as follows. These include the following references:
Broadmeadow and Nisbet [10], Smiley et al. [13], Goeller et al. [43], Lizotte et al. [63], Zedler [104], Benbow et al. [105], Lowe and LaLiberte [106], Hauer and Resh [107], McGarvey et al. [108], Carter et al. [109] , and Simon and Evans [110]
Lines 309-328 revised manuscript: The revised text is as follows ‘…Ecological functional responses include physical processes of hydrology, morphology, and physical and chemical components that affect ecosystem responses [104-110]. Aspects of biogeochemistry including uptake, transport, and storage of nutrients are also important elements of lotic and lentic aquatic ecosystems that can assist with understanding ecosystem responses to agricultural conservation practices [10,43,63,104]. To date, ecological research in agricultural watersheds has encompassed a broad suite of ecological responses scaling along the hierarchy of biological organization including nitrogen biogeochemistry, microbial ecology of phytoplankton and heterotrophic bacteria, a multitude of ecotoxicological assessments ranging from standard toxicity bioassays to biomarker assessments, and fish ecology assessments involving population and community ecology (see: Figure 6a-b). However, most studies are conducted using comparisons of watersheds across a gradient of CA and/or stressor conditions, such as comparisons of ecological responses between a reference watershed and agriculturally impacted watershed…’
- Broadmeadow, S.; Nisbet, T.R. The effects of riparian forest management on the freshwater environment: a literature review of best management practice. Hydrol Earth Sys Sci 2004, 8, 286-305.
- Smiley, P.C., Gillespie, R.B. Influence of physical habitat and agricultural contaminants on fishes in agricultural drainage ditches. Pages 37-73 in Agricultural Drainage Ditches: Mitigation Wetlands of the 21st Century, M. Moore and R. Kroger (editors). Research Signpost, Kerula, India. 2010.
- Goeller, B.C.; Febria, C.M.; McKergow, L.A.; Harding, J.S. Combining tools from edge-of-field to in-stream to attenuate reactive nitrogen along small agricultural waterways. Water 2020, 12, 383. doi.org/10.3390/w12020383
- Lizotte, R.E.; Shields, F.D.; Murdock, J.N.; Knight, S.S. Responses of Hyalella azteca and phytoplankton to a simulated agri-cultural runoff event in a managed backwater wetland. Chemosphere 2012, 87, 684–691. doi.org/10.1016/j.chemosphere.2011.12.058
- Zedler, J.B. Wetlands at your service: reducing impacts of agriculture at the watershed scale. Front Ecol Environ 2003, 1, 65–72.
- Benbow, M.E.; Pechal, J.L.; Ward, A.K. Heterotrophic bacteria production and microbial community assessment. In Methods in Stream Ecology Volume 1: Ecosystem Structure, 3rd ed.; Lamberti, G.A., Hauer, F.R., Eds.; Academic Press: San Diego, Cali-fornia, USA, 2017; Volume 1, pp. 161-176.
- Lowe, R.L; LaLiberte, G.D. Benthic stream algae: Distribution and structure. In Methods in Stream Ecology Volume 1: Ecosystem Structure, 3rd ed.; Lamberti, G.A., Hauer, F.R., Eds.; Academic Press: San Diego, California, USA, 2017; Volume 1, pp. 193-222.
- Hauer, F.R.; Resh, V.H. Macroinvertebrates. In Methods in Stream Ecology Volume 1: Ecosystem Structure, 3rd ed.; Lamberti, G.A., Hauer, F.R., Eds.; Academic Press: San Diego, California, USA, 2017; Volume 1, pp. 297-320.
- McGarvey, D.J.; Falke, J.A.; Li, H.W.; Li, J.L. Fish assemblages. In Methods in Stream Ecology Volume 1: Ecosystem Structure, 3rd ed.; Lamberti, G.A., Hauer, F.R., Eds.; Academic Press: San Diego, California, USA, 2017; Volume 1, pp. 321-354.
- Carter, J.L.; Resh, V.H.; Hannaford, M.J. Macroinvertebrates as biotic indicators of environmental quality. In Methods in Stream Ecology Volume 2: Ecosystem Function, 3rd ed.; Lamberti, G.A., Hauer, F.R., Eds.; Academic Press: San Diego, California, USA, 2017; Volume 2, pp. 293-318.
- Simon, T.P.; Evans, N.T. Environmental quality assessment using stream fishes. In Methods in Stream Ecology Volume 2: Eco-system Function, 3rd ed.; Lamberti, G.A., Hauer, F.R., Eds.; Academic Press: San Diego, California, USA, 2017; Volume 2, pp. 319-334.
In the Conclusion section you commented:
“Particularly, new research efforts involving the individual (e.g., genetic, biochemical, physiological) and population level (e.g., reproductive output) impacts of agricultural contaminants would assist with understanding the sublethal impacts of agricultural contaminants and the potential benefits of water quality improvements beyond the current regulatory standards”. However, you have provided too little data on this issue.
The authors agree that there is little data available addressing this issue and that is why, in the conclusions, the authors have suggested that future and new research efforts need to focus on these specific ecological endpoints. The authors cannot provide additional citations and references for research that is needed on this issue and that is the justification for stating that more work needs to be done to improve our understanding. The reviewer’s comment is, from one perspective, a tautology since the reviewer is reinforcing the point the authors are making in the conclusion statement. Because there is a paucity of information on this issue, the issue needs further study and trying to search for additional references to demonstrate that there is little information on this issue is no likely to be successful. For these reasons, the authors cannot provide what the reviewer is asking for.
Novel research that documents the influence of CAs on ecosystem structure and function within and among watersheds would be useful to further our understanding of whether conservation practices can restore aquatic ecosystems. This is an interesting approach. Please add more in the Discussion section.
As suggested by the reviewer, the authors have expanded upon the concept of novel research approaches as follows.
Lines 588-604 revised manuscript: added ‘…One example would be expanding existing research tools and techniques to assess nutrient dynamics from multiple stacked or integrated CAs to runoff to downstream receiving waterbodies using geographic information system framework with empirically measured denitrification rates across the landscape and into streams, rivers and lakes to map hot spots of nitrogen transport and transformation. Coinciding with this would be assessments of HABs nutrient limitation and nutrient sensitivities to determine how nitrogen affects HABs and possible nitrogen thresholds needed to mitigate HABs. Another example would be to expand the above efforts to include carbon cycling and aquatic system metabolism using state-of-the-art dissolved oxygen monitoring and models across an empirically measured (e.g., area in ha) gradient of CAs within or among multiple watersheds. A third example would be utilizing and expanding research networks across disciplines (engineering, biology, soil science, agronomy, social science) and multiple watersheds at national and international scales. For example, USDA-ARS Long-Term Agroecosystem Research Network (LTAR; https://ltar.ars.usda.gov/) can provide research opportunities to examine the influence of CAs on ecosystem services at large (continental) scales and include scientists from government, academic, and private sectors…’
Closing these information gaps will provide landowners and stakeholders with better management plans and tools to more efficiently maximize the benefits and ecosystem services of implementing CAs within agricultural watersheds. Ecosystem services are a very broad topic, and not were found in the text. Why you have applied this issue in the Conclusion section? Please delete or add more to the review paper.
Lines 604-607 revised manuscript: The statement has been deleted from the paper as suggested by the reviewer.

Round 2
Reviewer 3 Report
The author has modified the manuscript as per my suggestion. I suggest accepting the manuscript in its present form.
Author Response
The authors thank the reviewer in helping the authors improve the manuscript for the journal Water.
Reviewer 4 Report
Thanks.
Well-written texts responding to fashionable water management and agriculture needs. You have reacted effectively to the recommendations and all my suggestions. Check all parts for text editing and linguistic correctness.
Author Response
The authors thank the reviewer for assisting the authors with improving the manuscript for the Journal Water. The authors have reviewed the manuscript for text editing and linguistic correctness. The 2nd revised manuscript is provided.